



# Comparative Study of Strongly and Weakly Coupled Soil Moisture Data Assimilation with a Global Coupled Land-Atmosphere Model

Kenta Kurosawa[1,2], Shunji Kotsuki[1,3,4,5,6], Takemasa Miyoshi[1,2,5,6,7]

[1] RIKEN Center for Computational Science, Kobe, Japan
[2] Department of Atmospheric and Oceanic Science, University of Maryland, College Park,  Maryland, USA
[3] Center for Center for Environmental Remote Sensing, Chiba University, Chiba, Japan
[4] PRESTO, Japan Science and Technology Agency, Chiba, Japan
[5] RIKEN interdisciplinary Theoretical and Mathematical Sciences Program, Kobe, Japan
[6] RIKEN Cluster for Pioneering Research, Kobe, Japan
[7] Japan Agency for Marine-Earth Science and Technology, Yokohama, Japan

*Correspondence to*: Kenta Kurosawa (kkurosaw@umd.edu), Shunji Kotsuki (shunji.kotsuki@chiba-u.jp)

**Abstract.** This study explores coupled land-atmosphere data assimilation (DA) for improving weather and hydrological forecasts by assimilating soil moisture (SM) data. To assimilate land data with a coupled land-atmosphere model, weakly-
coupled DA has been a common approach, in which land (atmospheric) data are not used to analyze atmospheric (land) model variables.  This study integrates a land DA component into a global atmospheric DA system of the Nonhydrostatic ICosahedral Atmospheric Model (NICAM) and the Local Ensemble Transform Kalman Filter (LETKF), so that we can perform strongly-coupled land-atmosphere DA experiments. We perform various types of coupled DA experiments by assimilating atmospheric observations and SM data simultaneously. The results show that analyzing atmospheric variables
by assimilating SM data improves SM analysis and forecasts and mitigates a warm temperature bias in the lower troposphere where a dry SM bias exists. However, analyzing SM by assimilating atmospheric observations has detrimental impacts on SM analysis and forecasts.

## 1 Introduction

The Earth's natural environment can be considered a unified system in which several subsystems (e.g., atmosphere,
hydrosphere, cryosphere, and biosphere) interact with each other. Coupled models consider at least two of the Earth's subsystems and have been developed to emulate such interactions within unified systems. For example, coupled land–atmosphere models consider land–atmosphere interactions by passing the output data from the land subsystem to the atmospheric subsystem and vice versa during model time integrations. Coupled models represent more realistic physical processes and provide improved predictions of Earth's phenomena compared to those that consist of only a single
component.

Data assimilation (DA) plays an important role in numerical weather prediction (NWP) by providing accurate initial conditions. Some studies investigated coupled DA for ocean–atmosphere interactions (e.g., Zhang et al., 2007; Sugiura et al., 2008; Fujii et al., 2009; Frolov et al., 2016; Laloyaux et al., 2016; Sluka et al., 2016; Browne et al., 2019; Penny and Hamill



2017; Penny et al., 2019) and land-atmosphere interactions (e.g., de Rosnay et al., 2012; Lea et al., 2015; Suzuki et al., 2017;
Sawada et al., 2018; Draper and Reichle, 2019; Fairbairn et al., 2019).

In this study, we focus on experiments to evaluate the potential benefits of assimilating synthetic soil moisture (SM) data
from the Global Land Data Assimilation System (GLDAS; Rodell et al., 2004), within a controlled experimental setup
through the effective use of land-atmosphere interactions via data assimilation. Specifically, this study investigates whether
assimilating atmospheric (land) observational data into land (atmospheric) models is beneficial for their subsequent
forecasts. We employ SM data from GLDAS, a comprehensive and reliable dataset which facilitates simple data handling
and is suitable and sufficient for this study (cf. Section 2d). SM is particularly important among land variables because it
controls the exchange of water and energy between the atmosphere and land surface (Bateni and Entekhabi, 2012). For
example, SM has a profound impact on the evolution of boundary layers and precipitation during the warm season, when
there is high incoming radiation and evapotranspiration (Betts, 2009; Dirmeyer and Halder, 2016; Drusch and Viterbo,
2007). Moreover, improving SM data is essential for enhancing seasonal-scale climate predictions (Dirmeyer, 2000;
Douville and Chauvin, 2000; Drusch, 2007; Hauser et al., 2017). With a regional NWP system, Santanello et al. (2019)
showed that SM DA changed surface fluxes, evolution, and entrainment of the planetary boundary layer, and ambient
weather.

Two well-known coupled DA methods are weakly coupled DA and strongly coupled DA (cf. section 2.b). As one argument,
Lawless (2012) noted that strongly coupled DA is preferable for environmental prediction, as discussed at the 2012
International Workshop on Coupled Data Assimilation. A follow-up workshop in Toulouse in 2016 further elaborated on the
need for coupled DA. As for ocean-atmosphere models, Penny et al. (2019) explored a method to improve the initialization
process using a simplified model. They estimated ocean conditions with atmospheric observations and vice versa, and found
strongly coupled DA approaches were generally superior to weakly coupled approaches when using the simple toy model.
As Tang et al. (2021) states, however, regarding more complex models, it is unclear whether strongly coupled DA generally
outperforms weakly coupled DA. When it comes to land-atmosphere models, several studies have demonstrated the benefits
of strongly DA approaches for medium-range NWP (Suzuki et al., 2017; Sawada et al., 2018). In terms of assimilation of
land observations, while weakly coupled land–atmosphere DA is still the mainstream in NWP systems (e.g., Zhang et al.,
2007; Lea et al., 2015; Draper and Reichle, 2019), several studies have already examined the benefits of strongly coupled
DA on land observations. For example, Lin and Pu (2019, 2020) assimilated surface SM, 2-m temperature and humidity, and
conventional atmospheric observations, showing advantages of strongly coupled DA. They also showed that SM had crucial
impacts on the temperature field rather than the other variables. Thus, it is already known that SM DA is beneficial for the
coupled land-atmosphere models, but updates of cross-components have not yet been explored enough. Therefore, this study
aims at exploring better strategies to assimilate SM data in a strongly coupled land-atmosphere DA system.

This study uses a global atmospheric DA system known as the NICAM-LETKF (Terasaki et al., 2015), which comprises the
Nonhydrostatic Icosahedral Atmospheric Model (NICAM; Satoh et al., 2008, 2014) and the Local Ensemble Transform
Kalman Filter (LETKF; Hunt et al., 2007). NICAM incorporates the Minimal Advanced Treatments of Surface Interaction
and RunOff model (MATSIRO; Takata et al., 2003) as the land surface subsystem. We implement coupled land–atmosphere
DA in NICAM-LETKF to assimilate SM observations using either the weakly or strongly coupled DA methods. Our primary
scientific question is whether the assimilation of synthetic observational data from one model into another can improve
compatibility between the two models in the NICAM-LETKF system. In addition to conventional atmospheric observations
and AMSU-A radiances in NICAM-LETKF, this study assimilates SM data as hydrological observations.

This article is organized as follows. Section 2 describes the newly developed coupled land–atmosphere DA system. The
experimental settings are described in section 3. The results are presented and discussed in section 4. Finally, a summary is
provided in section 5.



## 2 Methodology

### 2.1 NICAM and MATSIRO models

NICAM is an icosahedral-grid-based atmospheric model that has been widely used for NWP (e.g., Kotsuki et al., 2019b, 2019c) and climate-scale predictions (e.g., Kodama et al., 2015; Kikuchi et al., 2017). We use NICAM with a 112-km horizontal resolution and 38 vertical layers to a height of approximately 40 km. Due to the relatively coarse horizontal resolution, the Arakawa and Schubert scheme (Arakawa and Schubert, 1974) and Berry's parameterization (Berry, 1967) are employed for cumulus parameterization and the large-scale condensation scheme, respectively. See Satoh et al. (2008) and
Satoh et al. (2014) for further details about NICAM.

MATSIRO represents all the major processes of water and energy exchange between land and atmosphere. MATSIRO consists of five vertical layers used for simulating soil temperature and moisture: 0−0.05, 0.05−0.25, 0.25−0.5, 0.5−0.75, and 0.75−2 meters. Surface energy and water fluxes are computed from their budgets at the ground and canopy surfaces in snow-free and snow-covered regions, considering the subgrid-scale snow distribution (Takata et al., 2003). SM is calculated in
each soil layer; SM is representative of the entire land component of a model grid area, whether it is snow-covered or not. Note that, in general, SM in NWP models has been updated using 2-m temperature and humidity observations for decades (e.g. Mahfouf et al., 2000; de Rosnay et al., 2014; Gomez et al., 2020).

### 2.2 LETKF and coupled data assimilation implementations

LETKF is a type of ensemble Kalman filter (EnKF; Evensen, 2003) that has been used for atmospheric, hydrological, and
oceanic DA. LETKF solves the analysis equations at every model grid point by assimilating the subset of observations within its localization influence radius. The analysis equations of LETKF are based on the ensemble transform Kalman filter (Bishop et al., 2001):

$$\bar{\mathbf{x}}^a = \bar{\mathbf{x}}^f + \delta\mathbf{X}^f\bar{\mathbf{w}}^a, \tag{1}$$

$$\bar{\mathbf{w}}^a = \widetilde{\mathbf{P}}^a\big(\mathbf{H}\delta\mathbf{X}^f\big)^{\mathrm{T}}\mathbf{R}^{-1}\big(\mathbf{y}^o - \mathbf{H}\bar{\mathbf{x}}^f\big), \tag{2}$$

$$\delta\mathbf{X}^a = \delta\mathbf{X}^f\mathbf{W}^a, \tag{3}$$

$$\delta\mathbf{X}^a = \delta\mathbf{X}^f[(m-1)\widetilde{\mathbf{P}}^a]^{\frac{1}{2}}, \tag{4}$$

where $\bar{\mathbf{x}}$ is the ensemble-mean model state, $\delta X$ is the ensemble perturbation matrix, H is the linear observation operator, R is the observation error covariance matrix, y is the observation data, and $\widetilde{\mathbf{P}}^a$ is the model state error covariance matrix in ensemble space, while superscript letters $a$, $f$, and $o$ denote analysis (posterior), forecast (prior), and observation,
respectively. Here, P is used for the error covariance in model space, and $\widetilde{\mathbf{P}}$ is used for the error covariance in the ensemble space. m is the ensemble size. $\bar{\mathbf{w}}$ is the ($m \times 1$) ensemble transform vector for the ensemble mean updates, and W is the ($m \times m$) ensemble transform matrix for ensemble perturbation updates. The analysis error covariance matrix $\widetilde{\mathbf{P}}^a$ is given by

$$\widetilde{\mathbf{P}}^a = [(m-1)\mathbf{I} + (\mathbf{H}\delta\mathbf{X}^f)^{\mathrm{T}}\mathbf{R}^{-1}\mathbf{H}\delta\mathbf{X}^f]^{-1}, \tag{5}$$

where I is the identity matrix. In practice, since the error covariance matrix $\widetilde{\mathbf{P}}^a$ is often underestimated, and filters eventually
become unstable, the introduction of the model error or variance inflation is necessary for stable filtering. The theoretical explanation of the model error can partially be attributed to the model nonlinearity under the perfect model assumption. In



this study, instead of adding random noise as the model error, we use a relaxation method at the end of the DA process, as described in section 3.

The analysis equation of the ensemble mean (Eqs. 1 and 2) is equivalent to the original analysis equation of the Kalman filter:

$$\bar{\mathbf{x}}^a = \bar{\mathbf{x}}^f + \delta\mathbf{X}^f \widetilde{\mathbf{P}}^a \left(\mathbf{H}\delta\mathbf{X}^f\right)^{\mathbf{T}} \mathbf{R}^{-1} \left(\mathbf{y}^o - \mathbf{H}\bar{\mathbf{x}}^f\right)$$
$$= \bar{\mathbf{x}}^f + \mathbf{P}^f \mathbf{H}^{\mathbf{T}} \left(\mathbf{H}\mathbf{P}^f\mathbf{H}^{\mathbf{T}} + \mathbf{R}\right)^{-1} \left(\mathbf{y}^o - \mathbf{H}\bar{\mathbf{x}}^f\right). \tag{6}$$

Here, $\mathbf{P}^f$ is the model state error covariance matrix in model space. The EnKF uses an ensemble-based approximation to the
forecast error covariance:

$$\mathbf{P}^f \approx \frac{1}{m-1} \delta\mathbf{X}^f (\delta\mathbf{X}^f)^{\mathbf{T}}. \tag{7}$$

Since the observation operator is linear in this study, for coupled land–atmosphere models, $\mathbf{P}^f$ is given by

$$\mathbf{P}^f = \begin{pmatrix} \left(\mathbf{P}^f\right)_{\mathbf{AA}} & \left(\mathbf{P}^f\right)_{\mathbf{AL}} \\ \left(\mathbf{P}^f\right)_{\mathbf{LA}} & \left(\mathbf{P}^f\right)_{\mathbf{LL}} \end{pmatrix}, \tag{8}$$

where A and L represents the atmosphere and land, respectively. For example, AA represents the atmospheric model and
atmospheric observations, and LA represents the land model and atmospheric observations. In practice, since some observations have nonlinear observation operators, the following approximation is required:

$$\mathbf{H}\delta\mathbf{X}^f \approx H\left(\bar{\mathbf{x}}^f \mathbf{1}^{\mathbf{T}} + \delta\mathbf{X}^f\right) - \overline{H\left(\bar{\mathbf{x}}^f \mathbf{1}^{\mathbf{T}} + \delta\mathbf{X}^f\right)},$$
(9)

where $H$ is the nonlinear observation operator, and 1 denotes a column vector with all $m$ elements being equal to 1. For
coupled models, $\mathbf{P}^f$ is approximated by

$$\left(\mathbf{P}^f\right)_{\alpha\beta} \approx \frac{1}{m-1} \delta\mathbf{X}^f_\alpha (\delta\mathbf{X}^f_\beta)^{\mathbf{T}}, \tag{10}$$

where $\boldsymbol{\alpha}$ and $\boldsymbol{\beta}$ are atmospheric or land variables (i.e., A or L).

For the weakly coupled DA (hereafter, WCDA) method, atmospheric observations are used only for updating NICAM state variables, and land observations for those of MATSIRO (Figure 1 a). That is, the cross-component error covariance between
atmospheric and land variables is assumed to be 0 in WCDA (i.e., $\left(\mathbf{P}^f\right)_{\mathbf{AL}} = \mathbf{0}$ and $\left(\mathbf{P}^f\right)_{\mathbf{LA}} = \mathbf{0}$). Thus, impacts of atmospheric observations can propagate to land model states, and vice versa, only through interactions between NICAM and MATSIRO during model forecasts. For the strongly coupled DA (hereafter, SCDA) method, the cross-component covariance is estimated based on ensemble forecasts (i.e., $\left(\mathbf{P}^f\right)_{\mathbf{AL}} \neq \mathbf{0}$, $\left(\mathbf{P}^f\right)_{\mathbf{LA}} \neq \mathbf{0}$ ,or both are nonzero). Therefore, atmospheric or land observations are used to update both NICAM and MATSIRO variables based on the cross-component
covariance (Figure 1 b). SCDA extracts more information than WCDA from the same observations if an appropriate forecast error covariance $\left(\mathbf{P}^f\right)_{\alpha\beta}$ is applied.



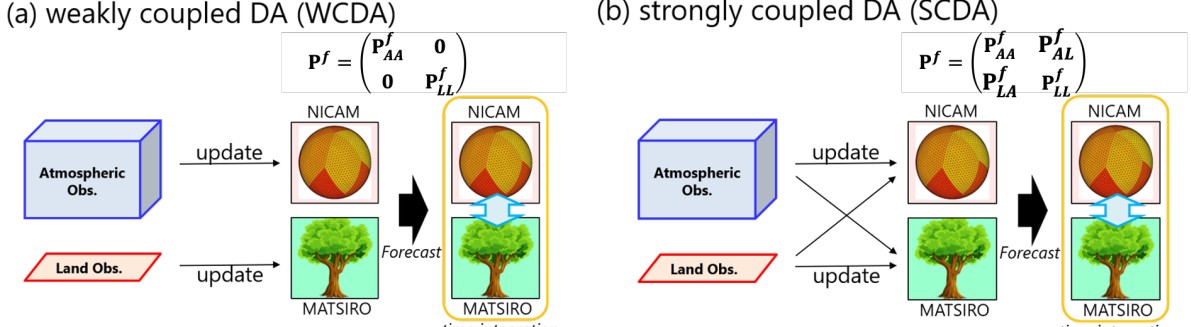

**Figure 1: Schematic images of (a) weakly coupled and (b) strongly coupled land–atmosphere data assimilation (DA) methods. Thin black arrows indicate model state updates through DA. Cyan double-headed arrows indicate land–atmosphere interactions**
**between NICAM and MATSIRO during subsequent model forecasts. Here, panel (b) shows the full strongly coupled DA method (cf. panel g of Figure 2). The image for NICAM was adapted from Satoh et al. (2014).**



This study considers seven coupled DA experiments (Figure 2), including WCDA and SCDA. Here, we introduce IDs indicating which observation type is assimilated for each model. First of all, this study defines that A and L represent the atmospheric and land models, respectively. Then, we set IDs: $A_-^\times$ means the atmospheric model assimilating only atmospheric observations, $A_\times^\times$ means the atmospheric model assimilating both atmospheric and land observations, $L_-^\times$ means the land model assimilating only atmospheric observations, $L_\times^-$ means the land model assimilating only land observations, $L_\times^\times$ means the land model assimilating both atmospheric and land observations, and $L_-^-$ means the land model assimilating no observation. For example, in $A_-^\times L_-^-$, NICAM variables are updated only by atmospheric data, whereas no land observations are assimilated for the land model (Figure 2 a). This experiment is equivalent to the standard NICAM-LETKF system without SM DA, or the control case (hereafter CTRL). $A_-^\times L_\times^-$ represents WCDA (Figures 2 b), $A_\times^\times L_\times^\times$ represents SCDA (Figures 2 g), and the other four experiments are quasi-strongly coupled DA, or one-way strongly coupled DA (hereafter "one-way SCDA") methods (Figures 2 c-f).

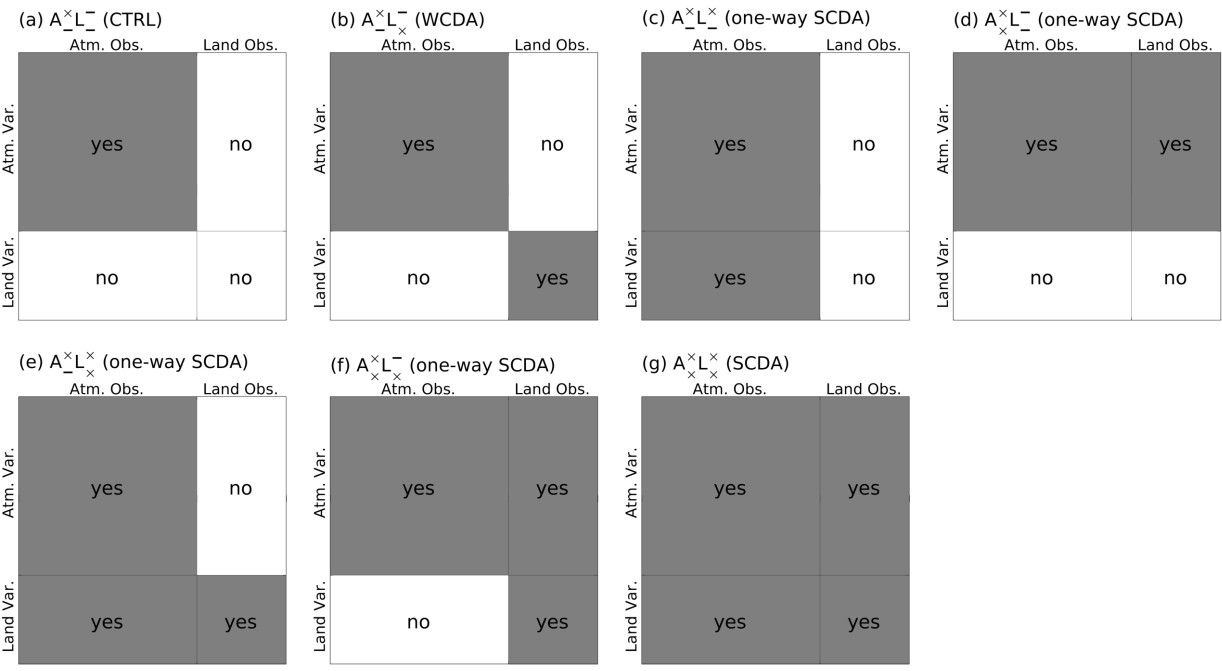

**Figure 2: Schematic plots of seven DA experiments for (a) $A_-^\times L_-^-$ (CTRL), (b) $A_-^\times L_\times^-$ (WCDA), (c) $A_-^\times L_-^\times$ (one-way SCDA), (d) $A_\times^\times L_-^-$ (one-way SCDA), (e) $A_-^\times L_\times^\times$ (one-way SCDA), (f) $A_\times^\times L_\times^-$ (one-way SCDA), and (g) $A_\times^\times L_\times^\times$ (SCDA). The vertical axis represents atmospheric or land variables, and the horizontal axis shows observations. The shading of variables matches that of the observations used for their updates. White areas with 'no' indicate error correlations that are assumed to be zero in DA. Gray areas with 'yes' indicate error correlations that are included in DA. Panel (b) indicates the weakly coupled DA (Figure 1 a). Panels (c–g) represent strongly coupled DA with cross-error correlations between atmospheric variables and land observations, as well as between land variables and atmospheric observations. Among the strongly coupled DAs, panel (g) indicates the full strongly coupled DA (Figure 1 b).**





This study designs specific configurations of one-way SCDA to investigate whether updating MATSIRO variables through assimilating particular atmospheric observations has a beneficial impact. This investigation aims at finding the best-performing coupled land-atmosphere DA that consists of updates with a beneficial effect for the experimental setting of the
present study. The best-performing one-way SCDA might be different if we use different DA configurations or change the experimental settings, such as resolution and DA frequency.



## 2.3 Atmospheric data

The original NICAM-LETKF system assimilates conventional observations from the NCEP operational system (a.k.a. NCEP
PREPBUFR), satellite radiance from Advanced Microwave Sounding Unit-A (AMSU-A), and the near-real-time version of
Global Satellite Mapping of Precipitation (GSMaP_NRT). The data set includes a number of different types of data:
radiosondes, wind profilers, aircraft reports, surface pressure, atmospheric motion vectors (AMVs) and surface winds
derived from satellite observations. The channel selections for satellite radiances are 6, 7, and 8 for AMSU-A. The
stratospheric sensitive channels are not assimilated in this study, considering their low altitude of 40 km above the model.
The satellite radiance's scan and airmass bias are adaptively estimated and corrected at each data assimilation cycle. This
experimental setting followed the operationally running NICAM-LETKF system.  In this study, we use these data as
atmospheric observations (cf. Table 1 of Kotsuki et al., 2019a). For further details of the assimilation methods used for these
observations, we refer readers to previous studies (Terasaki et al., 2015; Kotsuki et al., 2017a; Terasaki and Miyoshi, 2017).

## 2.4 Soil moisture data

Satellite instruments can measure several land variables, including SM, surface skin temperature, and snow depth. Previous
studies have found that land surface models tend to overestimate SM relative to SM data derived from satellite observations
(Bindlish et al., 2018). GLDAS also showed larger SM values than satellite-based data (Bi et al., 2016). The significant bias
between the model-based estimate and observation is unfavorable for DA. Prior to DA experiments, we compare spatial
distributions of climatological SM for NICAM and satellite-based observations from the Soil Moisture and Ocean Salinity
(SMOS; https://smos-diss.eo.esa.int/oads/access/) and the Advanced Microwave Scanning Radiometer 2 of Global Change
Observation Mission – Water (GCOMW/AMSR-2; https://lance.nsstc.nasa.gov/amsr2-science/). We can see that NICAM
SM is greatly biased compared to these satellite-based data (Figure 3 a, c, and d). In contrast, the bias of NICAM's SM
relative to GLDAS is much smaller than that relative to SMOS and GCOMW/AMSR-2.

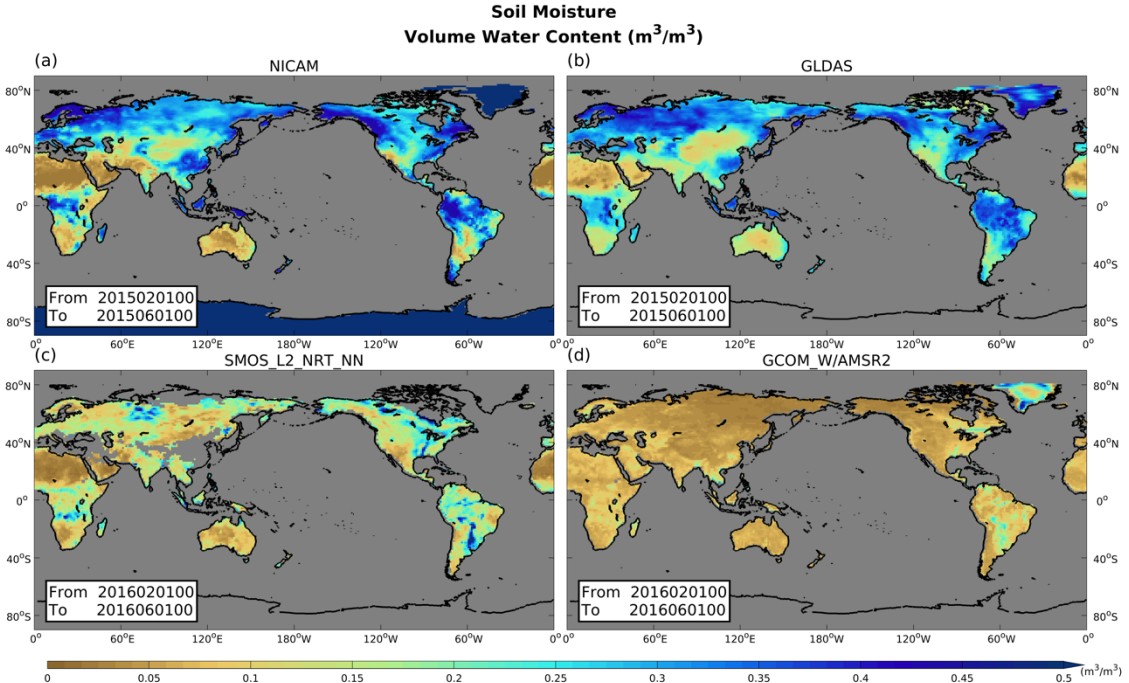





**Figure 3: Spatial patterns of soil moisture for NICAM, GLDAS, SMOS_L2_NRT_NN, and GCOM_W/AMSR2, averaged over February to June in 2015 (a, b) and 2016 (c, d).**



Hoover and Langland (2017) assimilated pseudo-radiosonde observations from an independent atmospheric reanalysis system. They mentioned that assimilating reanalysis data from this advanced system significantly reduced biases in atmospheric temperature and height. As a first step, this study takes a similar approach and assimilates SM from GLDAS to 200 avoid using satellite observation data which usually contain significant bias.

It is generally known that satellite, remote sensing, and model data sets have different average SM values. Since we do not know the true mean values in remote sensing or model outputs, we cannot attribute these differences in mean values to bias in a particular data source. Satellite retrieval and model averages are determined by the parameters used in the retrieval and surface models, but we also do not know what those parameters should be. Therefore, the standard approach in SM data 205 assimilation is to remove the difference between modeled and observed SM averages, and then assimilate only the temporal anomalies in the observed SM values. Since it is crucial to have unbiased model and observation states to ensure the DA assumption is correct, several processes are proposed (Dee, 2005). For example, Reichle and Koster (2004) suggest a simple method to remove strong biases between satellite-based and model-based data, in which they match the cumulative distribution functions (CDF) of the satellite and model data (a.k.a. CDF matching approach). On the other hand, several 210 previous studies have successfully performed data assimilation without bias correction (e.g., De Lannoy et al., 2007; Bosilovich et al., 2007; Reichle et al., 2010; Honda et al., 2018). For example, Honda et al. (2018) demonstrated that assimilating geostationary satellite infrared radiance observations without bias correction every 10 minutes reduced the bias between the forecast and observations, leading to improved analysis without causing inconsistencies in the model states. Following the success of these previous studies, the present study assimilates SM data without bias correction. As shown 215 later in Section 4a, the bias between the forecast and observation becomes negligible after a one-month spin-up period when SM from GLDAS is assimilated every 6 hours. Consequently, assimilating SM data without bias correction yields improvements in prediction accuracy of atmospheric variables. Since employing bias correction techniques and assimilating real satellite-sensed SM data could potentially lead to further enhancements, such endeavors are important subjects for future studies.

220 We perform QC using flags provided with the satellite observation data. In addition, as applied for PREPBUFR and GSMaP_NRT observations, we simply apply a gross error check for SM in which observations are rejected when the observation-minus-forecast value is greater than 10 times the observation error standard deviation (Terasaki et al. 2015).

GLDAS is a research-oriented land surface reanalysis system that produces spatiotemporally continuous global SM data. Among GLDAS datasets, this study uses Noah model-based SM data (GLDAS Noah Land Surface Model L4 Version 2.1; 225 Chen et al., 1996; Koren et al., 1999). We assimilate only first-layer SM since satellite measurements cannot observe deep-layer SM. GLDAS provides 3-hourly SM at a spatial resolution of 0.25° × 0.25°. As these data are denser than those of NICAM (112-km and 6-hourly resolution), we reduced the data density spatially and temporally. The original SM data are averaged within a NICAM model grid so that each observation corresponds to one model grid point. The original 3-hourly data are also averaged over 6 hours. These spatial and temporal data aggregation processes were carried out simultaneously 230 prior to assimilation.

The GLDAS Version 2.1 simulation is forced with National Oceanic and Atmospheric Administration (NOAA)/ Global Data Assimilation System (GDAS) atmospheric analysis fields (Derber et al., 1991), the disaggregated Global Precipitation Climatology Project (GPCP) V1.3 Daily Analysis precipitation fields, and the Air Force Weather Agency's AGRicultural METeorological modeling system (AGRMET) radiation fields. Because GLDAS uses observed precipitation of GPCP, 235 GLDAS' SM is considered better than that of MATSIRO, which uses NICAM's precipitation forecasts to drive the land surface model. Since NICAM' SM has a large bias against the satellite-based product (Figure 3), this study assimilates GLDAS' SM as pseudo-observations as Hoover and Langland (2017) and verifies forecasted SM compared to GLDAS.

The detection depth of actual observations, such as microwave sensors, is much shallower than the GLDAS soil depth of 10 cm. The present experimental setting for assimilating GLDAS SM data may result in more significant impacts than the 240 experiments with actual satellite observation intervals.



## 3. Experimental setting

This study performs 40-member NICAM-LETKF experiments. NICAM ensemble forecasts are performed for 9-hour intervals, and observation data from the last 6-hour period are assimilated. The initial ensemble members of the experiments were obtained from the 1st-40th members of a long-term 128-member NICAM-LETKF experiment (Terasaki et al. 2019).
This means the initial ensemble spread of SM relies on initial conditions perturbed by the ensemble NICAM forecasts. Covariance localization in LETKF is applied to the observation error covariance R so that distant observations have smaller impacts on the analysis (Hunt et al., 2007; Miyoshi and Yamane, 2007). Gaussian functions are used for horizontal and vertical localization, given by:

$$f = \exp\left[-\frac{1}{2}\{(d_h/\sigma_h)^2 + (d_v/\sigma_v)^2\}\right],$$
250       (11)

where $f$ is the localization function and $d_h$ and $d_v$ are the horizontal distance (km) and vertical difference (log(Pa)) between the analysis model grid point and the observation, respectively. Standard deviations (SDs) $\sigma_h$ and $\sigma_v$ are 400 km and 0.4 natural log pressure as Terasaki et al. (2019) implemented. The localization function is replaced by zero beyond $2\sqrt{10/3} \cdot \sigma_{h,v}$. Land (atmospheric) observations are assimilated into the atmospheric (land) model using the same vertical localization
scale. For land observations, surface pressure (Ps) is assigned for the observed height. This study uses relaxation to prior spread (RTPS; Whitaker and Hamill, 2012) for covariance inflation. For atmospheric variables, the relaxation parameter is set to 0.90, which is determined through sensitivity tests (Kotsuki et al., 2017b). The original NICAM-LETKF method, which assimilates only atmospheric observations, is referred to as the control experiment. These experimental settings have been widely applied in previous NICAM-LETKF experiments (e.g., Kotsuki et al., 2018, 2019a). In addition to atmospheric
observations, this study assimilates SM data as hydrological observations. The observation error SD of SM is estimated at 0.05 (m³ m⁻³) based on the innovation statistics of Desroziers (2005) (cf. appendix A). We perform one control experiment and six SM DA experiments, as shown in the schematic images of Figure 2.

Maintaining the ensemble spread is important in the EnKF. We initially expected that ensemble forecasts could sufficiently keep the ensemble spreads of MATSIRO variables due to physical coupling with NICAM. However, the ensemble spread of
MATSIRO's SM decreased rapidly after initiating assimilation of SM in our preliminary experiment (not shown). We did not solve this rapid reduction of ensemble spreads even by applying RTPS with relaxation parameter $\alpha$=0.90. This outcome seems to be related to two fundamental challenges: (1) the land models are typically more dependent on external forcing, rather than being modeled as a chaotic dynamical system dependent on initial conditions, and (2) the timescales of dynamical changes in land models are much longer than those in the atmospheric models. The latter implies that the land
model is likely to have a long memory beyond 6 hours for SM. In the case of assimilating SM with atmosphere-land coupled models, SM observations correspond to the slow mode, and atmospheric variables correspond to the fast mode. Therefore, offline land DA systems usually inflate the ensemble spread by adding random noise to atmospheric forcing or observational data. For example, Reichle et al. (2002) added perturbations to the ensemble system, specifically to forcing and to the model states variables, to account for sources of model error in the land model forecast to generate an ensemble representative of
the model forecast uncertainty. In the current study, we use RTPS to maintain the ensemble spread of MATSIRO's SM to avoid the ensemble becoming too confident. In addition, land DA experiments with the land-atmosphere system would represent model errors to some extent since each land model is driven by different forcing. The relaxation parameter for SM is set to $\alpha$=1.00 so that the analysis ensemble spread is equivalent to the forecast ensemble spread. For further details on creating ensemble spreads for land models, we encourage readers to review the summary presented in Draper (2021).

Further, since satellite-borne microwave sensors can measure only surface layer SM, this study explores better DA strategies that will be applicable to satellite observations. Thus, this study only assimilates surface-layer SM by land DA, and SM data only in the surface layer (0cm-10cm) provided by GLDAS is used. Although it can be expected that analyzing deeper layer



SM would also be essential to take advantage of land-atmosphere coupling, this study focuses on the surface layer (0cm-10cm) where feedbacks to the atmosphere would be more pronounced than deeper layers.

We first performed a spin-up NICAM-LETKF experiment from June to September 2014 by assimilating only atmospheric observations. The initial NICAM ensemble conditions are taken from the long-term NICAM-LETKF experiment of Terasaki et al. (2019). DA experiments are performed for 13 months, from 0000 UTC 1 October 2014 to 1800 UTC 30 November 2015. The first month (October 2014) is considered as a spin-up period, and the results for the latter 12 months are used for validation.

We assimilate SM from GLDAS, and in Section 4a, the data are used for validation to check if the assimilation behaves as expected (i.e., the analysis departures of SM are reduced by the assimilation). In addition, we also use SM from ERA5 reanalysis data (Hersbach et al., 2020) as an independent dataset for validation scores.

We evaluate atmospheric variables against the ERA5 reanalysis data in Section 4b. The analysis of land variables is performed separately from the atmospheric analysis in the ERA5 by assimilating screen-level temperature, dewpoint, and
synoptic observations with the optimal interpolation. While the ERA5 assimilates no SM observation, the ERA5 assimilates many more satellite observations than the NICAM-LETKF, such as Microwave Humidity Sounder and Advanced Technology Microwave Sounder. Therefore, validating NICAM-LETKF atmospheric fields relative to the ERA5 is reasonable. Furthermore, as described, SM of GLDAS can be considered better than the NICAM-LETKF because it is derived by observed precipitation. Hence, in the following sections, we show that the assimilation of GLDAS SM has a
reasonably beneficial effect on atmospheric fields of NICAM-LETKF, verified by comparison with ERA5.

## 4. Results and discussion

### 4.1 Impacts on soil moisture

We first examine the impacts of SM assimilation on MATSIRO. Figure 4 compares the global bias patterns for the prior state of SM at the near-surface layer (i.e., 0-5cm) relative to GLDAS, averaged over 12-months period from November 2014
to October 2015. Three panels show the results for $\mathbf{A}_-^\times \mathbf{L}_-^-$ (CTRL), $\mathbf{A}_-^\times \mathbf{L}_\times^-$ (WCDA), and $\mathbf{A}_\times^\times \mathbf{L}_\times^\times$ (SCDA). $\mathbf{A}_-^\times \mathbf{L}_-^-$ (CTRL) shows dry biases relative to GLDAS in general, especially in the continent of Africa, South America, Australia, and Central Eurasia (Figure 4 a). Assimilating SM into MATSIRO successfully mitigates these SM biases (Figures 4 b and c). Furthermore, assimilating SM mitigates the wet SM bias in regions where SM is overestimated with $\mathbf{A}_-^\times \mathbf{L}_-^-$ (CTRL). Therefore, the newly developed coupled land–atmospheric DA system successfully assimilates SM data into MATSIRO, and
we confirm the developed DA system works well. These results are expected and not surprising because forecasts are validated using the same data as observations. No notable differences are observed in global bias patterns between $\mathbf{A}_-^\times \mathbf{L}_\times^-$ (WCDA) and $\mathbf{A}_\times^\times \mathbf{L}_\times^\times$ (SCDA) in global bias patterns (Figures 4 b and c).









**Figure 4: Global patterns of 6-hour forecast bias for soil moisture (SM; m³ m⁻³) relative to GLDAS for (a) A$_-^\times$L$_-^-$ (CTRL), (b) A$_-^\times$L$_\times^-$**
**(WCDA), and (c) A$_\times^\times$L$_\times^\times$ (SCDA) averaged over 12 months from November 2014 to October 2015. The blue and brown colors represent overestimated and underestimated SM values relative to GLDAS, respectively.**

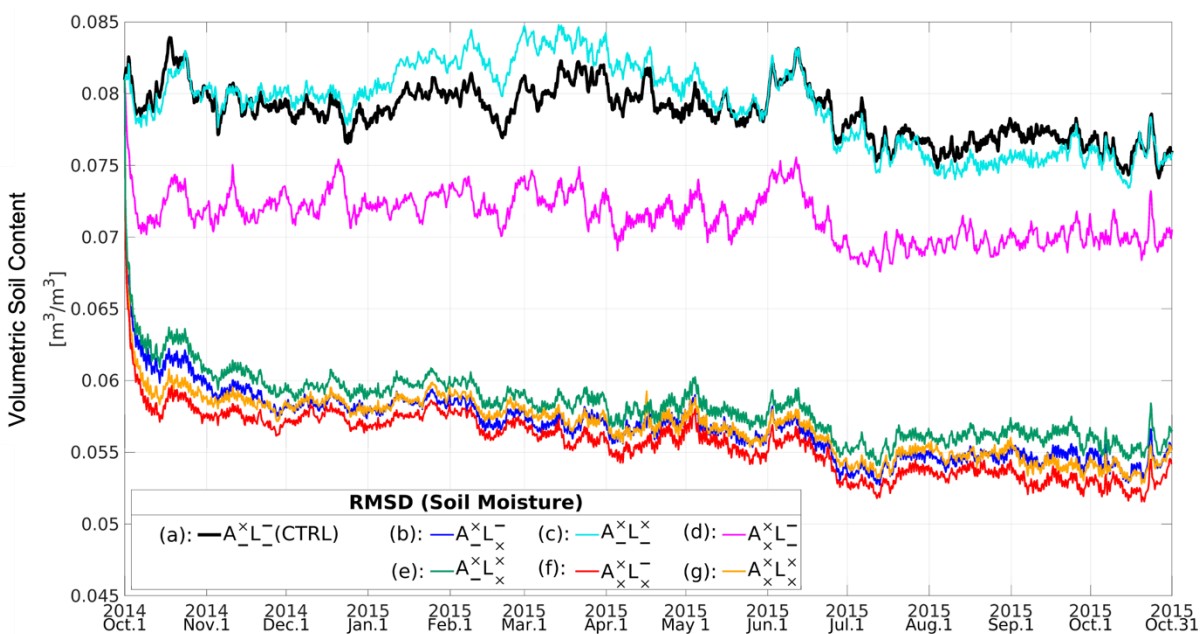

**Figure 5: Time series of global-mean forecast root mean square differences (RMSDs) for soil moisture (SM; m³ m⁻³) relative to GLDAS. The black, blue, cyan, magenta, green, red, and yellow lines indicate the (a) A$_-^\times$L$_-^-$ (CTRL), (b) A$_-^\times$L$_\times^-$ (WCDA), (c) A$_-^\times$L$_-^\times$ (one-way SCDA), (d) A$_\times^\times$L$_-^-$ (one-way SCDA), (e) A$_-^\times$L$_\times^\times$ (one-way SCDA), (f) A$_\times^\times$L$_\times^-$ (one-way SCDA), and (g) A$_\times^\times$L$_\times^\times$ (SCDA) experiments, respectively. Experiments (a−g) correspond to the DA patterns (a−g) shown in Figure 2.**

Figure 5 shows the time series of global-mean root mean square differences (RMSDs) for SM relative to GLDAS. All experiments that assimilate SM have smaller errors in SM than in A$_-^\times$L$_-^-$ (CTRL). Although A$_-^\times$L$_\times^-$ (WCDA) and A$_\times^\times$L$_\times^\times$ (SCDA) show reduced errors, no clear difference is apparent between the two experiments. Among the seven experiments, A$_\times^\times$L$_\times^-$ (one-way SCDA) results in the smallest SM error. In A$_\times^\times$L$_\times^-$ (one-way SCDA), SM observations are used for updating both NICAM and MATSIRO, whereas atmospheric observations are used only for updating NICAM and not for MATSIRO.
Since A$_\times^\times$L$_\times^-$ (one-way SCDA) results in better SM estimation than A$_\times^\times$L$_\times^\times$ (SCDA), updating SM in MATSIRO through assimilation of atmospheric observations may have detrimental impacts on SM in the experimental settings of this study. Such detrimental impacts are also found by comparing other cases, such as A$_-^\times$L$_-^\times$ (one-way SCDA) and A$_-^\times$L$_\times^-$ (WCDA). The larger error in A$_-^\times$L$_-^\times$ (one-way SCDA) than in A$_-^\times$L$_\times^-$ (WCDA) probably comes from erroneous error covariance estimates between atmospheric observations and land variables ($\left(\mathbf{P}^f\right)_{\mathbf{AL}}$) due to insufficient ensemble size. Ensemble-based DA can
provide spurious error correlations when the ensemble size is small. Assimilating observations based on spurious error covariances generally degrades the analysis results (cf. variable localization of Kang et al. 2011). Moreover, the difference in timescale between the atmospheric and terrestrial models may have a dominant influence, which could be verified by experiments using a short assimilation window. Such further investigation of the assimilation window is essential for future



studies of land-atmosphere coupled DA. $\mathbf{A}^\times_\times \mathbf{L}^\times_-$ (one-way SCDA) shows similar RMSDs to that of $\mathbf{A}^\times_- \mathbf{L}^-_-$ (CTRL), which
implies that atmospheric observations have neither positive nor negative impacts on SM. Because many types of atmospheric observations are assimilated in this study, clarifying impacts of individual observation type is complicated. The results might be changed if we assimilate only one kind of atmospheric observation, such as precipitation data, with the variable localization. Accurate estimation of $\left(\mathbf{P}^f\right)_{\mathbf{AL}}$ by increasing the number of ensembles might reduce the RMSD of $\mathbf{A}^\times_\times \mathbf{L}^\times_-$. Penny et al. (2019) also faced this kind of problem when assimilating slower ocean observation data into an atmosphere-ocean
model with coupled DA. Penny et al. (2019) found that it was more difficult to use slow-mode observations (from the ocean) to update the fast-mode (atmosphere). They overcame this problem by using larger ensembles and increasing the analysis update and observation frequency. As discussed for maintaining ensemble spreads for SM, SM observations correspond to the slow mode and atmospheric variables correspond to the fast mode in our experimental settings. Therefore, applying Penny et al. (2019)'s approach may further improve SCDA.

$\mathbf{A}^\times_\times \mathbf{L}^-_\times$ (one-way SCDA) shows smaller SM errors than $\mathbf{A}^\times_\times \mathbf{L}^-_-$ (WCDA). Also, $\mathbf{A}^\times_\times \mathbf{L}^-_-$ (one-way SCDA) shows smaller SM errors than $\mathbf{A}^\times_- \mathbf{L}^-_-$ (CTRL). Therefore, updating NICAM variables by assimilating SM observations is consistently beneficial to SM analysis. Also, as already shown, updating SM in MATSIRO by assimilating the atmospheric observations has detrimental impacts on SM. Therefore, the error correlation between the SM observations and the atmospheric model variables is more reliable than that between the atmospheric observations and the land model's SM variable.

In terms of reducing the errors in SM, the optimal coupled DA method in our experimental setting was $\mathbf{A}^\times_\times \mathbf{L}^-_\times$ (one-way SCDA). The errors in SM can be reduced by updating atmospheric and land variables through the assimilation of SM. Several previous studies have found that it is important to correct the "upstream" dynamics in the coupled system (e.g., by Sluka et al., 2016). In other words, since the atmosphere strongly drives the land via surface forcing, correcting the atmospheric variables would improve forecasts of the coupled land surface model. From the point of view of the land model,
the SM can be updated accurately by assimilating the observed SM directly. Attempting to use fast-varying atmospheric observations for updating SM would lead to suboptimal analysis because of the non-perfect ensemble-based error covariance estimate between atmospheric observations and modelled SM. In contrast, the detrimental impacts of updating atmospheric variables by $\left(\mathbf{P}^f\right)_{\mathbf{AL}}$ cancel out the beneficial impacts of updating SM by $\left(\mathbf{P}^f\right)_{\mathbf{LA}}$. Therefore, for our model configuration and DA design, $\mathbf{A}^\times_\times \mathbf{L}^\times_\times$ (SCDA) is less effective than $\mathbf{A}^\times_- \mathbf{L}^-_\times$ (WCDA). This problem might happen because the DA approach
degrades the analysis when assimilating atmospheric data into the land model. The approaches for atmosphere-ocean coupled DA suggested by Penny et al. (2019) could solve the problem, which will be an important future subject to improve SCDA even more.

Figure 6 shows the ensemble spread of SM. Because RTPS is used with a relaxation parameter of 1.0 for land variables, the ensemble spread does not change during DA. The ensemble spreads are stable during the experimental period. Because no
significant difference is observed in the ensemble spreads among experiments, the difference in RMSDs relative to GLDAS must originate from the difference of the update strategy. The ensemble spread of $\mathbf{A}^\times_- \mathbf{L}^\times_-$ (one-way SCDA) is the smallest among these cases, which means the atmospheric observations have collapsed the spread more than any other configurations. By assimilating the atmospheric observations into the land model, the impact of the land observations becomes less, thus the detrimental effect in those cases. This could also be related to the balance between the errors on the atmospheric
observations and the spread of the land model variables, and how these are calibrated through normalization. This indicates that the atmospheric observation error should be inflated when applied to the land DA via SCDA without applying averaging. The process would filter out the impact of high variability in the atmosphere, similar to adding errors of representativeness in the spatial dimension.



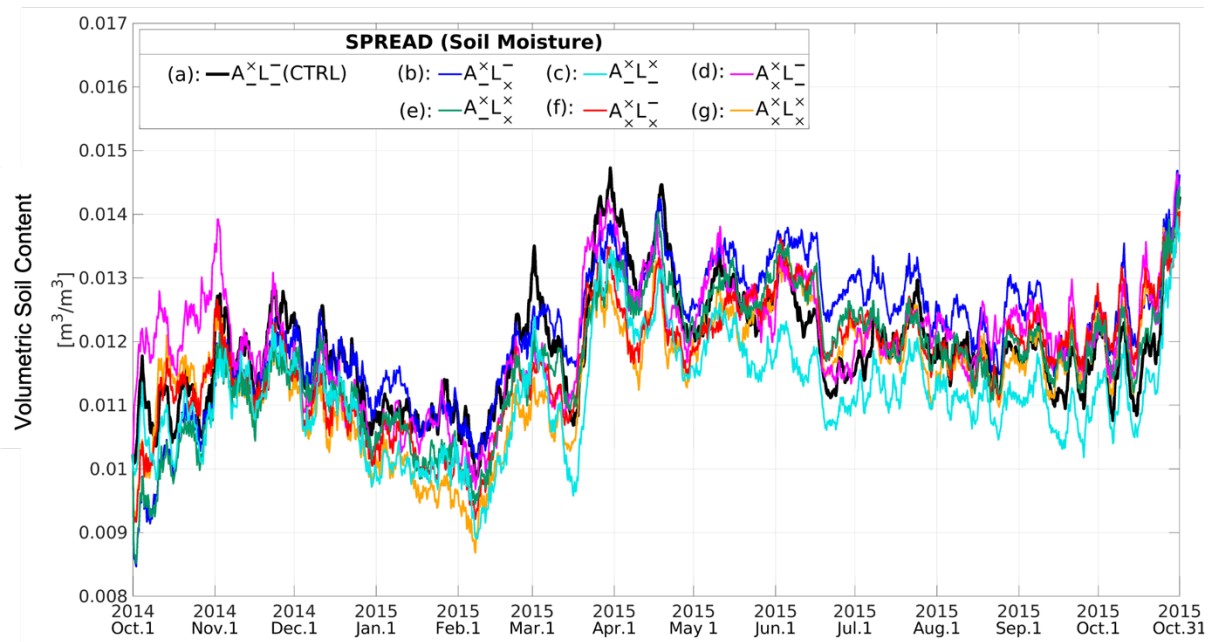

**Figure 6. Similar to Figure 5, but showing forecast ensemble spreads of SM (m³ m⁻³).**



Figure 7 shows the global patterns of differences in analysis RMSDs for SM, averaged over 12-months period from November 2014 to October 2015. We discuss three experiments: $\mathbf{A}_-^\times\mathbf{L}_\times^-$ (WCDA), $\mathbf{A}_\times^\times\mathbf{L}_\times^\times$ (SCDA), and $\mathbf{A}_\times^\times\mathbf{L}_\times^-$ (one-way

SCDA), which are the best three experiments in terms of errors in SM in Figure 4. First, we compare $\mathbf{A}_-^\times\mathbf{L}_\times^-$ (WCDA) and $\mathbf{A}_\times^\times\mathbf{L}_\times^-$ (one-way SCDA). Figure 7 (a) suggests that updating atmospheric variables with SM DA generally has beneficial impacts on SM. In South America, the Arabian Peninsula, and India, beneficial impacts are seen in regions where $\mathbf{A}_-^\times\mathbf{L}_-^-$ ( CTRL) shows a dry bias in SM. Additionally, beneficial impacts are apparent in Central Africa, where $\mathbf{A}_-^\times\mathbf{L}_-^-$ (CTRL) has a wet bias in SM. In contrast, SM DA has moderate impacts in North America and Eurasia. In these areas,

$\mathbf{A}_\times^\times\mathbf{L}_\times^\times$ (SCDA) performs worse than $\mathbf{A}_-^\times\mathbf{L}_\times^-$ (WCDA; Figure 7 b), suggesting that assimilating atmospheric observations to update SM in MATSIRO be detrimental in the experimental settings of this study. Therefore, eliminating the updates of MATSIRO with atmospheric observations has beneficial impacts for SCDA (Figure 7 c).





**Figure 7. Global patterns of soil moisture analysis RMSD (m³ m⁻³) relative to GLDAS averaged over 12 months from November 2014 to October 2015: (a) difference between $A_-^\times L_\times^-$ (WCDA) and $A_\times^\times L_\times^-$ (one-way SCDA), (b) difference between $A_-^\times L_\times^-$ (WCDA) and $A_\times^\times L_\times^\times$ (SCDA), and (c) difference between $A_\times^\times L_\times^\times$ (SCDA) and $A_\times^\times L_\times^-$ (one-way SCDA). Warm colors indicate that the latter experiments performing better than the former experiments, whereas cool colors indicate worse performance of the latter methods.**





We also investigate the SM correlations between GLDAS and the results of the experiments (Figure 8). We can see that the correlation to GLDAS is larger in the regions where positive impacts are observed in Figure 7. Figure 9 shows the results of the two-sample t-test. Time series of absolute bias of SM analysis relative to GLDAS is sampled from November 2014 to October 2015. When the P-values at a point are smaller than 5%, the null hypothesis at the 95% confidence level is rejected,

implying a significant difference. By the significance test, we can see the significant differences between each experiment over broad regions. The significant differences between methods $\mathbf{A}_-^\times \mathbf{L}_\times^-$ (WCDA) and $\mathbf{A}_\times^\times \mathbf{L}_\times^-$ (one-way SCDA) are mainly located in the areas where the bias was relatively substantial in Figure 4 a (Figure 9 a). From Figure 8 and Figure 9, we can reconfirm the points described in the comments about Fig. 7: (1) using SM to update atmospheric variables has positive effects, especially in areas where there are dry biases, (2) areas where there are wet biases are mitigated by SM DA, and (3)

updating SM with atmospheric observations has negative effects, leading to $\mathbf{A}_\times^\times \mathbf{L}_\times^\times$ (SCDA) being worse than the other two settings in our experiments.





**Figure 8. Global patterns of soil moisture analysis correlation relative to GLDAS averaged over 12 months from November 2014 to October 2015: (a) difference between $A_-^\times L_\times^-$ (WCDA) and $A_\times^\times L_\times^-$ (one-way SCDA), (b) difference between $A_-^\times L_\times^-$ (WCDA) and $A_\times^\times L_\times^\times$ (SCDA), and (c) difference between $A_\times^\times L_\times^\times$ (SCDA) and $A_\times^\times L_\times^-$ (one-way SCDA). Warm colors indicate that the latter experiments performing better than the former experiments, whereas cool colors indicate worse performance of the latter methods.**





**Figure 9. Global patterns of soil moisture analysis absolute bias (m³ m⁻³) relative to GLDAS: (a) difference between $A_-^\times L_\times^-$ (WCDA) and $A_\times^\times L_\times^-$ (one-way SCDA), (b) difference between $A_-^\times L_\times^-$ (WCDA) and $A_\times^\times L_\times^\times$ (SCDA), and (c) difference between $A_\times^\times L_\times^\times$ (SCDA) and $A_\times^\times L_\times^-$ (one-way SCDA). Only the areas where the T-test gives significant differences (the P-value < 5%) are colored, sampling with time series of soil moisture analysis from November 2014 to October 2015. Areas without significant differences are grayed out.**




Next, we also investigate the seasonal differences between $\mathbf{A}_-^{\times}\mathbf{L}_{\times}^-$ (WCDA) and $\mathbf{A}_{\times}^{\times}\mathbf{L}_{\times}^-$ (one-way SCDA). Figure 10 shows the difference of SM analysis RMSD relative to GLDAS between $\mathbf{A}_-^{\times}\mathbf{L}_{\times}^-$ (WCDA) and $\mathbf{A}_{\times}^{\times}\mathbf{L}_{\times}^-$ (one-way SCDA; Figure 10 a, c, e, g), and observed precipitation of GPCP version 1.3 (Figure 10 b, d, f, h). Over Central Africa, we can see that the peak of precipitation gradually shifts to the north from the south as time goes by (Figure 10 b, d, f, h). With the movement of the

peaks, the dominance of $\mathbf{A}_{\times}^{\times}\mathbf{L}_{\times}^-$ (one-way SCDA) over $\mathbf{A}_-^{\times}\mathbf{L}_{\times}^-$ (WCDA) is gradually becoming more pronounced. Furthermore, the precipitation over South America has been decreasing over time. With the decrease, the advantage of $\mathbf{A}_{\times}^{\times}\mathbf{L}_{\times}^-$ (one-way SCDA) over $\mathbf{A}_-^{\times}\mathbf{L}_{\times}^-$ (WCDA) is gradually being canceled out. In summary, we can see that updating atmospheric variables with SM has positive impacts, especially in regions and seasons with large amounts of precipitation.



**Difference in Mean RMSD; Soil Moisture [m³/m³]**
$\mathbf{A}_-^{\times}\mathbf{L}_{\times}^-$(WCDA) - $\mathbf{A}_{\times}^{\times}\mathbf{L}_{\times}^-$(one-way SCDA)

**Mean Precipitation [mm/6hr]; GPCP v1.3**




**Figure 10. Global patterns of soil moisture analysis RMSD (m$^3$ m$^{-3}$) relative to GLDAS (panels a, c, e, g) and spatial patterns of observed precipitation of GPCP version 1.3 (mm 6h$^{-1}$; panels b, d, f, h). Results are averaged over 3 months: (a, b) November 2014 to January 2015, (c, d) February to April 2015, (e, f) May to July 2015, and (g, h) August to October 2015. Panels (a, c, e, g) show the difference between $A_-^\times L_\times^-$ (WCDA) and $A_\times^\times L_\times^-$ (one-way SCDA). In panels (a, c, e, g), warm colors indicate that $A_\times^\times L_\times^-$ (one-way**

**SCDA) performing better than $A_-^\times L_\times^-$ (WCDA), whereas cool colors indicate worse performance of $A_\times^\times L_\times^-$ (one-way SCDA).**



We use ERA5 SM as an independent dataset for the validation scores, although so far, we have been using GLDAS to verify that the experimental setup works as expected. Figure 11 compares the global bias patterns for the prior state of SM at near-surface layer 5 as in Figure 4, but relative to ERA5. We can see that $\mathbf{A}_-^\times\mathbf{L}_-^-$ (CTRL) shows large dry biases relative to ERA5

in South America and Central Eurasia (Figure 11 a). The dry biases appear mitigated by updating MATSIRO with the SM of GLDAS. Furthermore, NICAM has a large dry bias in the center of the African continent relative to ERA5, unlike when compared to GLDAS in Figure 4. There is a wet bias at the southern and northern ends of the African continent, which increases with the assimilation of SM, but the global-averaged scores show improvements compared to $\mathbf{A}_-^\times\mathbf{L}_-^-$ (CTRL; Figures 11 b and c). No notable differences are observed between $\mathbf{A}_-^\times\mathbf{L}_\times^-$ (WCDA) and $\mathbf{A}_\times^\times\mathbf{L}_\times^\times$ (SCDA) in global bias patterns

(Figures 11 b and c).

Figure 12 shows the time series of global-mean RMSDs for SM as in Figure 5, but relative to ERA5. Similar to the results in Figure 5, we can find the following features: all experiments that assimilate SM have smaller errors in SM than in $\mathbf{A}_\times^\times\mathbf{L}_-^-$ (CTRL); $\mathbf{A}_-^\times\mathbf{L}_-^\times$ (one-way SCDA) shows similar RMSDs to that of $\mathbf{A}_\times^\times\mathbf{L}_-^-$ (CTRL); $\mathbf{A}_\times^\times\mathbf{L}_-^-$ (one-way SCDA) shows smaller SM errors than $\mathbf{A}_\times^\times\mathbf{L}_-^-$ (CTRL). The differences between the other four experiments, in which SM observations

update the MATSIRO variables, are unclear, but they show a significant decrease in RMSDs compared to $\mathbf{A}_\times^\times\mathbf{L}_-^-$ (CTRL).

Lastly, Figure 13 shows the differences between experiments for RMSD relative to ERA5. We can see a meaningful benefit of having atmospheric model variables updated by SM observations where there was a robust dry bias, e.g., in the South American continent (Figures 13 a and b). On the other hand, there was originally a wet bias against ERA5, i.e., the Arabian Peninsula and North of the African continent, resulting in a modification effect. Furthermore, a feature not seen in Figure 7

is that with ERA5 as reference data, there is no significant worsening of the MATSIRO variables by updating them with atmospheric observations (Figure 13 c).

The validation results using an independent dataset confirm that the experiments conducted in this study are functioning reasonably well. These findings support the notion that our experiments, which assimilate SM data from GLDAS without bias correction, can perform satisfactorily without violating the underlying assumptions of data assimilation. In this section,

the results showed that the assimilation of atmospheric observations could lead to adverse effects on soil moisture analysis. It is crucial to note that this issue stems from the experimental setup rather than statistical aspects. The primary cause of these adverse effects would be the weak dynamical relationship between the lower troposphere and SM. We will explore the issues related to this physical relationship in the subsequent section.



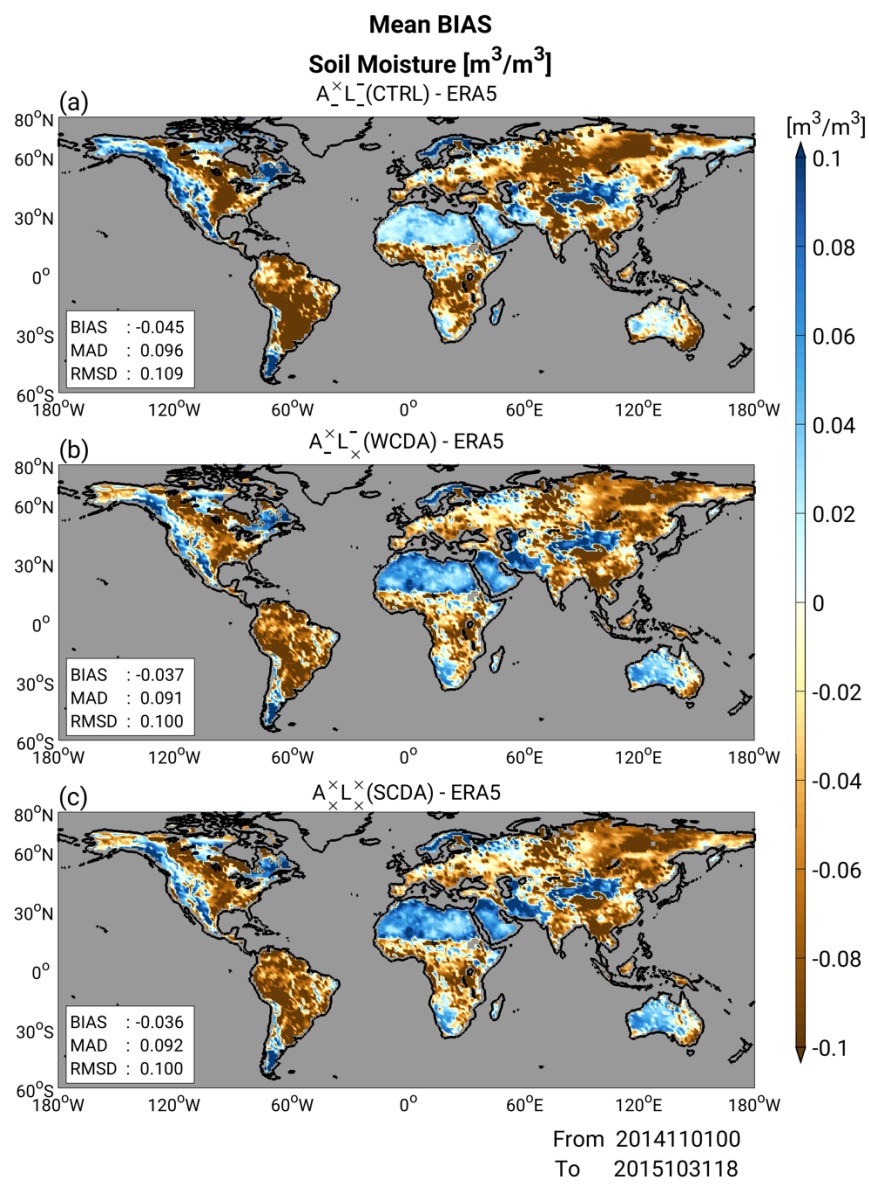


**Figure 11. Similar to Figure 4, but showing 6-hour forecast bias for soil moisture relative to ERA5 (m³ m⁻³).**



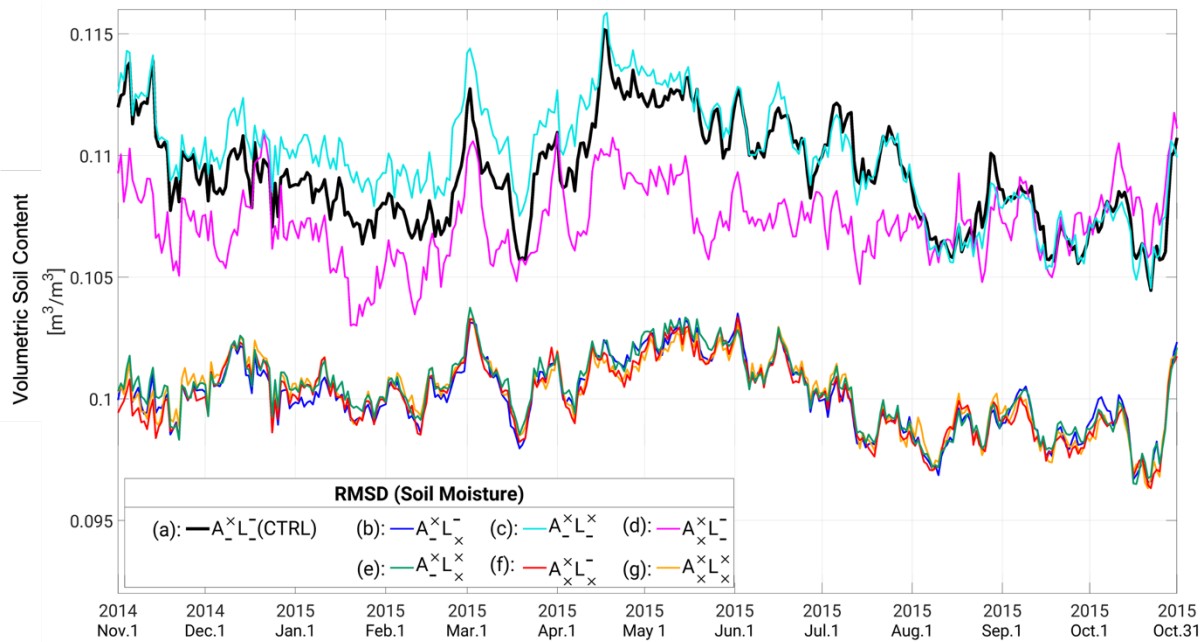

**Figure 12. Similar to Figure 5, but showing RMSDs for soil moisture relative to ERA5 (m³ m⁻³).**



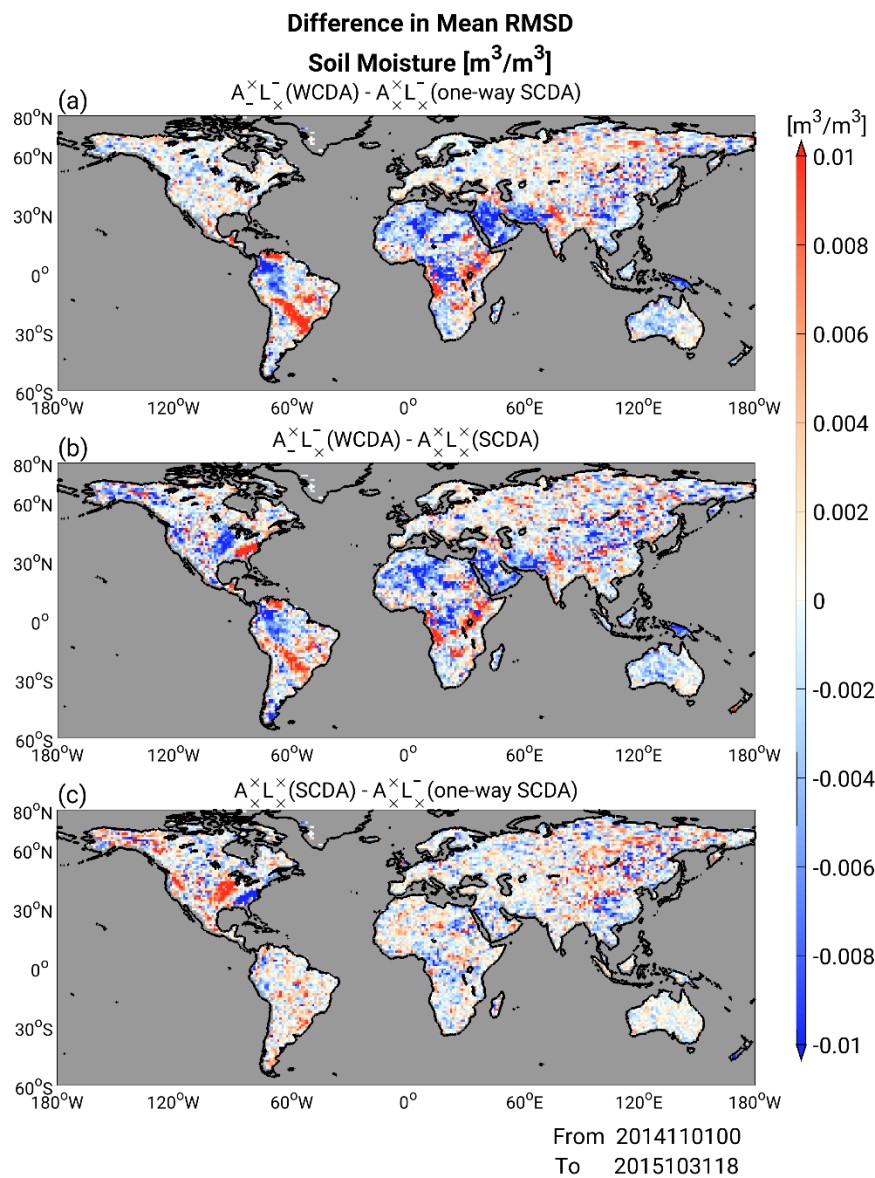

**Figure 13. Similar to Figure 7, but showing Global patterns of soil moisture analysis RMSD relative to ERA5 (m³ m⁻³).**




## 4.2 Impacts on atmospheric field

Here, we investigate the impacts of assimilation of SM on atmospheric variables. Figure 14 shows the global patterns of forecast biases for temperature (K) in the lower troposphere (850 hPa) relative to the ERA5 reanalysis data averaged over 12 months from November 2014 to October 2015. Hereafter, we discuss the results of $\mathbf{A}_{-}^{\times}\mathbf{L}_{-}^{-}$ (CTRL) and three coupled DA

experiments: $\mathbf{A}_{-}^{\times}\mathbf{L}_{\times}^{-}$ (WCDA), $\mathbf{A}_{\times}^{\times}\mathbf{L}_{\times}^{-}$ (one-way SCDA), and $\mathbf{A}_{\times}^{\times}\mathbf{L}_{\times}^{\times}$ (SCDA). Figure 14 (a) shows that $\mathbf{A}_{-}^{\times}\mathbf{L}_{-}^{-}$ (CTRL) has a warm temperature bias in regions with dry SM biases, as illustrated in Figure 4 (e.g., South America, Africa, and Australia). In these regions, increasing SM values after assimilation of SM decreases temperature estimates in the lower troposphere (Figures 14 b–d), since more of the incoming solar and longwave radiation is converted to latent heat flux, and less to sensible heat flux with greater SM. Compared to $\mathbf{A}_{-}^{\times}\mathbf{L}_{\times}^{-}$ (WCDA), however, $\mathbf{A}_{\times}^{\times}\mathbf{L}_{\times}^{-}$ (one-way SCDA) and $\mathbf{A}_{\times}^{\times}\mathbf{L}_{\times}^{\times}$ (SCDA)

show an overcooling effect for temperature in the continent of Africa and Australia (Figures 14 c and d). This overcooling effect is caused by the assimilation of SM into atmospheric variables in NICAM. The condition and type of soil determine the allocation of energy to latent and sensible heat flux. In areas with sufficient SM, evaporation is limited by the amount of available water, even though more evaporation is energetically possible. In such a case, the ratio of latent heat to sensible heat (i.e., Bowen ratio) will be determined by the surface temperature. In contrast, in a dry area, the ratio becomes smaller. In

addition, the energy balance is led by the turbulent fluxes of sensible, latent heat, and the ground heat flux. The energy transfer from the surface to the atmosphere creates spatial pressure gradients that drive atmospheric circulation at various scales. Due to the factors above, the most appropriate setting was $\mathbf{A}_{-}^{\times}\mathbf{L}_{\times}^{-}$ (WCDA) in our experiments. There are no remarkable changes in temperature over the ocean among the DA methods.

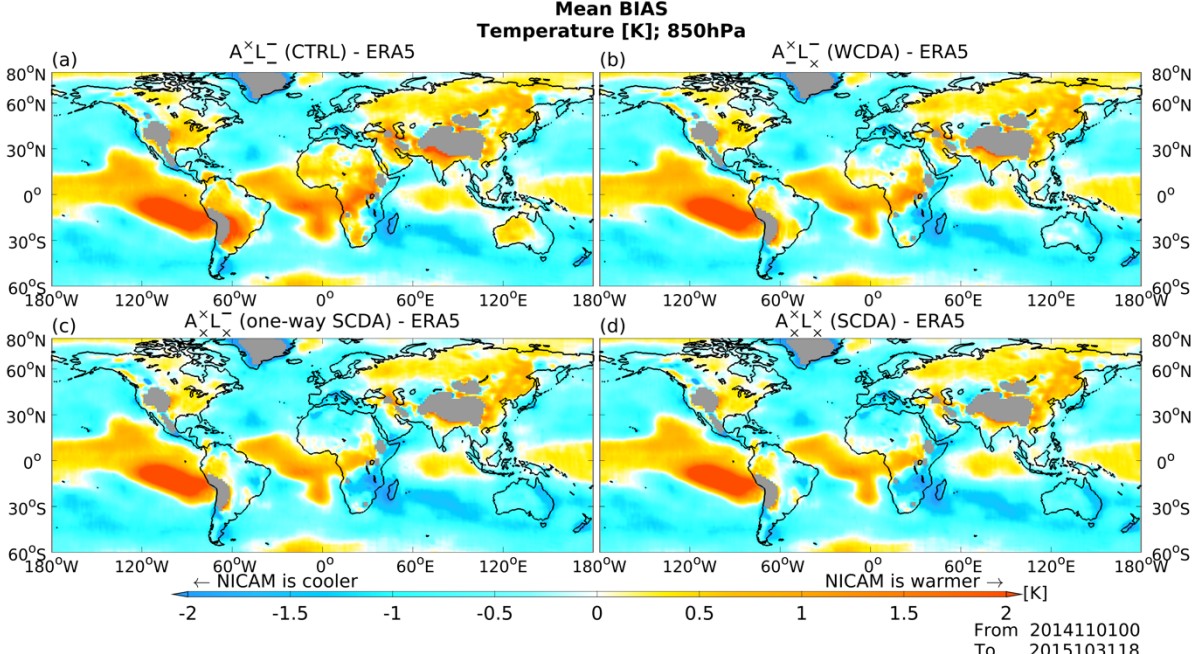

**Figure 14. Global patterns of forecast bias for temperature (K) at 850 hPa relative to ERA5 reanalysis values for (a) $\mathbf{A}_{-}^{\times}\mathbf{L}_{-}^{-}$ (CTRL), (b) $\mathbf{A}_{-}^{\times}\mathbf{L}_{\times}^{-}$ (WCDA), (c) $\mathbf{A}_{\times}^{\times}\mathbf{L}_{\times}^{-}$ (one-way SCDA), and (d) $\mathbf{A}_{\times}^{\times}\mathbf{L}_{\times}^{\times}$ (SCDA), averaged over 12 months from November 2014 to October 2015. Red and blue colors represent warm and cold biases, respectively.**




Table 1 summarizes the global-mean scores for bias, RMSD, and mean absolute difference (MAD) in temperature. Tables 1
(a) and (b) show these values averaged over the ocean and land, respectively. The errors in Table 1 (a) differ less strongly
than those in Table 1 (b), showing that assimilation of SM changes the temperature field mainly over land. The bias values in
Table 1 (b) show that $\mathbf{A^{\times}_{-}L^{-}_{-}}$ (CTRL) has a warm temperature bias over land in general. Assimilating SM has a cooling effect
and mitigates the warm temperature bias. However, $\mathbf{A^{\times}_{\times}L^{-}_{\times}}$ (one-way SCDA) and $\mathbf{A^{\times}_{\times}L^{\times}_{\times}}$ (SCDA) decrease temperature too
much, resulting in a cold bias. Consequently, $\mathbf{A^{\times}_{-}L^{-}_{\times}}$ (WCDA) results in the best temperature field among the four
experiments in terms of temperature bias at 850 hPa. Assimilating SM with $\mathbf{A^{\times}_{-}L^{-}_{\times}}$ (WCDA) decreases the average
temperature bias by 0.26 (K) over land. These changes over land do not propagate significantly to the temperature bias over
the ocean.

**Table 1. Averaged scores for bias, mean absolute difference (MAD), and RMSD for temperature at 850 hPa in Figure 14. The
biases and errors in (a) and (b) are averaged only over the ocean and only over land, respectively. The smallest errors are**
**indicated by the bold font.**

(a) over the ocean

| Temperature [K] | (i) $A^{\times}_{-}L^{-}_{-}$ (CTRL) | (ii) $A^{\times}_{-}L^{-}_{\times}$ (WCDA) | (iii) $A^{\times}_{\times}L^{-}_{\times}$ (one-way SCDA) | (iv) $A^{\times}_{\times}L^{\times}_{\times}$ (SCDA) |
|---|---|---|---|---|
| BIAS | **-0.352** | -0.382 | -0.434 | -0.443 |
| MAD | 1.366 | **1.363** | 1.379 | 1.375 |
| RMSD | 1.590 | **1.583** | 1.600 | 1.595 |

(b) over land

| Temperature [K] | (i) $A^{\times}_{-}L^{-}_{-}$ (CTRL) | (ii) $A^{\times}_{-}L^{-}_{\times}$ (WCDA) | (iii) $A^{\times}_{\times}L^{-}_{\times}$ (one-way SCDA) | (iv) $A^{\times}_{\times}L^{\times}_{\times}$ (SCDA) |
|---|---|---|---|---|
| BIAS | 0.200 | **-0.060** | -0.266 | -0.268 |
| MAD | 1.320 | **1.287** | 1.326 | 1.334 |
| RMSD | 1.564 | **1.510** | 1.544 | 1.555 |






We also investigate changes in the precipitation field. Here, we compare analysis increments and biases in precipitation data
over the continent of Africa, where large changes in SM due to SM DA occurred (Figure 4). Figures 15 (a−c) show the
spatial patterns of analysis increments for precipitation amount averaged over 12 months from November 2014 to October
2015. Note that DA can be used for analyzing not only model diagnosed variables (i.e., model state variables) but also other
model outputs. For example, Kotsuki et al. (2017a) analyzed precipitation using NICAM-LETKF, where precipitation is not
part of the initial condition. Here, we compare analysis increments of model-like precipitation (cf. Figure 3 of Kotsuki et al.,
2017a). Because we classified precipitation as an atmospheric diagnosed variable, precipitation analysis increments in
$\mathbf{A}_{-}^{\times}\mathbf{L}_{-}^{-}$ (CTRL) and $\mathbf{A}_{-}^{\times}\mathbf{L}_{\times}^{-}$ (WCDA) occur during assimilation of atmospheric observations. The difference in precipitation
analysis increments between $\mathbf{A}_{-}^{\times}\mathbf{L}_{-}^{-}$ (CTRL) and $\mathbf{A}_{-}^{\times}\mathbf{L}_{\times}^{-}$ (WCDA) is insignificant. In contrast, precipitation in $\mathbf{A}_{\times}^{\times}\mathbf{L}_{\times}^{-}$ (one-way
SCDA) can be affected by the assimilation of atmospheric and SM observations. Analysis increments shift markedly in
central Africa when precipitation is updated with SM DA (Figure 15 c). We observe negative analysis increments where
$\mathbf{A}_{-}^{\times}\mathbf{L}_{-}^{-}$ (CTRL)'s SM is drier and positive increments when it is wetter, suggesting that coupled land–atmospheric DA
perform reasonably. Namely, assimilating SM data increases (decreases) precipitation in areas where NICAM has a dry
(wet) bias (Figure 4 a).





**Figure 15. Spatial patterns of analysis increments for precipitation (mm 6h$^{-1}$; panels a−c), and precipitation forecast biases (panels d−f) and analysis biases (panels g−i) relative to GPCP version 1.3 (mm 6h$^{-1}$), averaged over 12 months from November 2014 to October 2015. Magenta and cyan colors in (a−c) represent increased and decreased precipitation with DA, respectively. The green and brown colors in (d−i) represent overestimated and underestimated precipitation values, respectively, relative to GPCP. Panels (a, d, g), (b, e, h), and (c, f, i) show the $A^{\cdot}_{\cdot}L^{-}_{\cdot}$ (CTRL), $A^{\cdot}_{\cdot}L^{-}_{\times}$ (WCDA), and $A^{\times}_{\times}L^{-}_{\times}$ (one-way SCDA) experiments, respectively.**



Spatial patterns of forecast and analysis biases in precipitation relative to Global Precipitation Climatology Project (GPCP) version 1.3 estimates are shown in Figures 15 (d)−(i). GPCP, which provides global precipitation data through the merging of various satellite and gauge datasets, is considered to include the best global precipitation estimates in the climate research community (Kotsuki et al., 2019c). First-guess precipitation in $\mathbf{A}^{\times}\mathbf{L}^{-}$ (CTRL) has a positive bias relative to GPCP (Figure 15 d; +0.159 mm 6h$^{-1}$), and this overestimation is intensified in $\mathbf{A}^{\times}\mathbf{L}^{-}_{\times}$ (WCDA; Figure 15 e; +0.184 mm 6h$^{-1}$). In contrast, first-
guess precipitation bias in $\mathbf{A}^{\times}_{\times}\mathbf{L}^{-}_{\times}$ (one-way SCDA); bias: +0.176 mm 6h$^{-1}$) is smaller than that in $\mathbf{A}^{\times}\mathbf{L}^{-}_{\times}$ (WCDA), although both experiments assimilate SM (Figures 15 e and f). In $\mathbf{A}^{\times}\mathbf{L}^{-}$ (CTRL) and $\mathbf{A}^{\times}\mathbf{L}^{-}_{\times}$ (WCDA), atmospheric variables are not updated through SM DA. Therefore, differences between the precipitation biases of forecasting and analysis occur due to assimilation of GSMaP_NRT in $\mathbf{A}^{\times}\mathbf{L}^{-}$ (CTRL) and $\mathbf{A}^{\times}\mathbf{L}^{-}_{\times}$ (WCDA). These two experiments result in differing precipitation biases due to biases in their precipitation forecasts (Figures 15 g and h). Assimilation of GSMaP_NRT slightly reduces the
bias in precipitation relative to GPCP (from 0.159 to 0.157 in $\mathbf{A}^{\times}\mathbf{L}^{-}$ (CTRL), and from 0.184 to 0.177 in $\mathbf{A}^{\times}\mathbf{L}^{-}_{\times}$ (WCDA)). In contrast, SM DA changes the analysis precipitation in $\mathbf{A}^{\times}\mathbf{L}^{-}_{\times}$ (WCDA). $\mathbf{A}^{\times}_{\times}\mathbf{L}^{-}_{\times}$ (one-way SCDA) shows the smallest bias in analysis precipitation. Therefore, estimating precipitation based on $\mathbf{A}^{\times}_{\times}\mathbf{L}^{-}_{\times}$ (one-way SCDA) during the analysis would be beneficial. That is, updating atmospheric variables with SM data plays an important role in improving the accuracy of precipitation. Compared to CTRL, one of the reasons for the larger bias in the $\mathbf{A}^{\times}\mathbf{L}^{-}_{\times}$ (WCDA) and $\mathbf{A}^{\times}_{\times}\mathbf{L}^{-}_{\times}$ (one-way SCDA)
is due to increased rainfall in areas where NICAM has the dry bias. Originally, NICAM overestimates precipitation (Kotsuki et al., 2019; Fig. 6). Improvements of soil moisture may have reinforced the bias, which leads to worse scores in those cases. It can be said that an improvement of the model bias contained in NICAM is necessary to solve this problem.

Figure 16 compares the forecast biases in precipitation relative to GPCP averaged over 3 months from June to August 2015. We selected this period to explore SM-atmosphere coupling, as suggested by Koster et al. (2004). Figure 16 (a) shows that
NICAM tends to overestimate precipitation in convergence regions at low latitudes (0°N−10°N) and underestimate precipitation in South America and Southeast and East Asia. Figures 16 (b) and (c) show changes in the precipitation forecasts of $\mathbf{A}^{\times}\mathbf{L}^{-}_{\times}$ (WCDA) and $\mathbf{A}^{\times}_{\times}\mathbf{L}^{-}$ (one-way SCDA). Assimilating SM affects precipitation mainly at low latitudes. Koster et al. (2004) found "hotspots" around the world where SM affects precipitation during June−August based on ensemble runs of global climate models. Koster et al. (2004) noted that the initial condition of SM was sensitive to rainfall
predictability over the North American Great Plains, equatorial Africa, and India (cf. Figure 1 of Koster et al., 2004). These areas correspond to the locations where forecast precipitation differed sharply from SM DA, as shown in Figures 16 (b) and (c), particularly for the Sahel, equatorial Africa, and India. Comparing $\mathbf{A}^{\times}_{\times}\mathbf{L}^{-}_{\times}$ (one-way SCDA) with $\mathbf{A}^{\times}\mathbf{L}^{-}_{\times}$ (WCDA), coupled DA shows stronger impacts in hotspots where the precipitation field is sensitive to the initial condition of SM.





**Mean FG BIAS**
**Precip. [mm/6hr]**



Figure 16. Spatial patterns of changes in precipitation (mm 6h⁻¹) averaged over 3 months from June to August 2014. Panels (a),
(b), and (c) show the difference between $A_\_^\times L_\_^-$ (CTRL) and GPCP, $A_\_^\times L_\_^-$ (CTRL) and $A_\_^\times L_\times^-$ (WCDA), and $A_\_^\times L_\_^-$ (CTRL) and
$A_\times^\times L_\times^-$ (one-way SCDA), respectively. The green and brown colors in (a) represent overestimated and underestimated precipitation
values relative to GPCP, and the red and blue colors in (b, c) represent increased and decreased precipitation values with SM DA,
respectively.






Figure 17 shows vertical cross-sections of forecast biases for temperature and vapor mixing ratio (Qv) relative to ERA5 reanalysis data along 20°E over the continent of Africa, averaged over 12 months from November 2014 to October 2015. $A_\times^\times L_-^-$ (CTRL) generally shows a warm temperature bias and a dry humidity bias near the land surface (1000−800 hPa). With the assimilation of SM, $A_\times^\times L_\times^-$ (WCDA) and $A_\times^\times L_\times^-$ (one-way SCDA) show decreases in temperature of the lower

troposphere at latitudes where $A_\times^\times L_-^-$ (CTRL) has a warm bias (Figures 17 b and c). Since the vertical layers of NICAM are almost the same as those of the ERA5, the cooling impacts would not be attributed to the difference in vertical resolutions between NICAM and ERA5. $A_\times^\times L_\times^-$ (WCDA) propagates the impacts of SM DA for atmospheric variables through the interaction between NICAM and MATSIRO during model time integrations. In addition, $A_\times^\times L_\times^-$ (one-way SCDA) updates atmospheric variables directly through SM DA. $A_\times^\times L_\times^-$ (one-way SCDA) alters atmospheric variables both directly and

indirectly. Therefore, $A_\times^\times L_\times^-$ (one-way SCDA) lowers temperature too much due to the strong interaction between SM and atmospheric variables (Figure 17 c). Figure 17 (d) shows that most land surface areas have dry Qv biases relative to the ERA5 reanalysis. However, $A_\times^\times L_-^-$ (CTRL) has a wet Qv bias around 5°N, perhaps due to the wet SM bias around this latitude (Figure 4 a). This result implies that the error in SM is directly linked with the error in Qv. $A_\times^\times L_\times^-$ (WCDA) and $A_\times^\times L_\times^-$ (one-way SCDA) correct for the bias caused by increased or decreased Qv near the surface using SM DA (Figures 17

e and f). $A_\times^\times L_\times^-$ (one-way SCDA) tends to correct Qv more strongly than $A_\times^\times L_\times^-$ (WCDA), as shown in Figure 17 (f). However, there are also strong corrections at the northern and southern edges of the cross sections, especially for Qv, which seem to deteriorate the model performance.

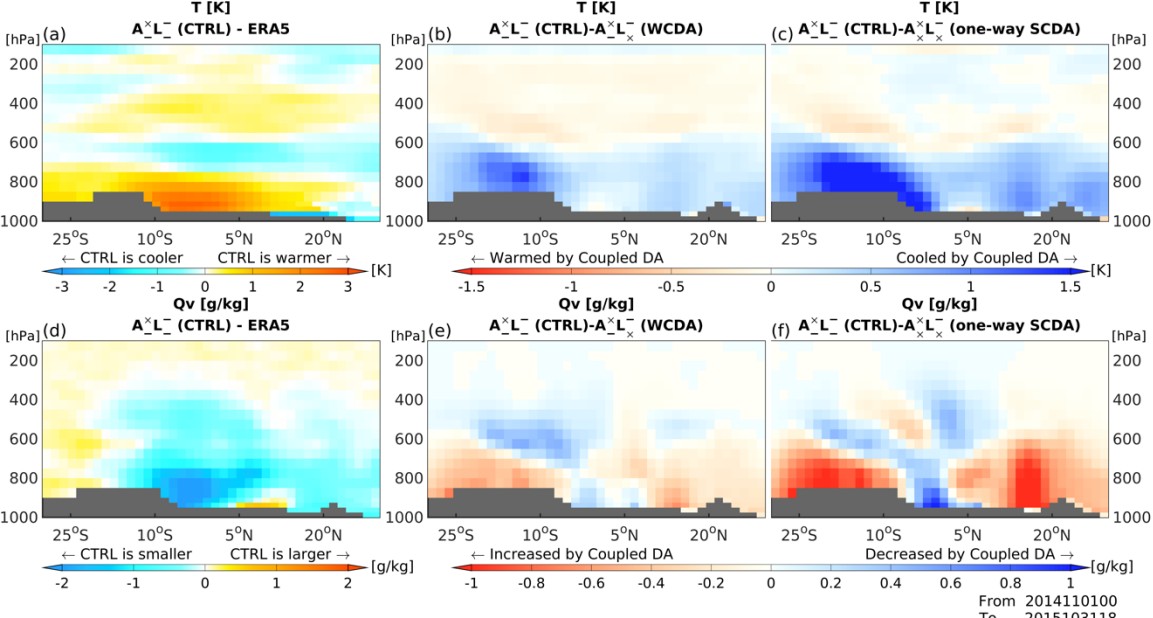

**Figure 17. Vertical cross-sectional plots of differences in (a−c) temperature (K) and (d−f) water vapor mixing ratio (g kg⁻¹)**
**averaged over 12 months from November 2014 to October 2015 along 20°E over the continent of Africa. Panels (a, d), (b, e), and (c, f) show the differences between $A_\times^\times L_-^-$ (CTRL) and the ERA5 reanalysis, $A_\times^\times L_-^-$ (CTRL) and $A_\times^\times L_\times^-$ (WCDA), and $A_\times^\times L_-^-$ (CTRL) and $A_\times^\times L_\times^-$ (one-way SCDA), respectively. The vertical and horizontal axes show the pressure level from 1000 to 100 hPa and the latitude, respectively.**



## 5. Conclusions


This study aimed to explore the optimal coupled land–atmospheric DA method for improving weather forecasts through the assimilation of hydrological observations. We implemented a coupled land–atmospheric DA into the NICAM-MATSIRO model and assimilated SM data from GLDAS. We performed a series of coupled DA experiments, including weakly and strongly coupled DA, and reached the following conclusions. Note that the assimilation of GLDAS pseudo soil moisture data

is not a realistic operational setting, as it is likely to have much better spatial and temporal coverage than real satellite observations.

The assimilation of SM successfully mitigated SM biases. Updating SM by assimilating atmospheric observations had detrimental impacts on SM, perhaps due to spurious error correlations between atmospheric observations and land model variables caused by insufficient ensemble size. In contrast, updating atmospheric model variables by assimilating SM

observations had beneficial impacts on SM, implying that the error correlation between SM observations and atmospheric model variables is more reliable. Consequently, the optimal coupled DA method in this study was $\mathbf{A}_\times^\times \mathbf{L}_\times^-$ (one-way SCDA), in which atmospheric and SM data were used to update the NICAM variables, but only SM data were used to update the MATSIRO variables. The results of this study indicate that $\mathbf{A}_\times^\times \mathbf{L}_\times^\times$ (SCDA) is less effective than weakly coupled DA, which may be caused by sampling errors and/or insufficient localization of the ensemble background-error covariance matrix. As

Penny et al. (2019) have shown, experiments with a simple model to examine several factors in detail, such as the number of ensemble members, the scale of localization, the spread of the ensemble of initial members, and the frequency of coupling intervals, would yield very important information. With adequate settings, such as proposed by Penny et al. (2019), the experiments with $\mathbf{A}_\times^\times \mathbf{L}_\times^\times$ (SCDA) might give a superior performance. In addition, the difference in dynamical timescales between the atmospheric and land models may possibly have a dominant influence. Using a shorter DA window with more

linear cross-domain dynamics could be useful to investigate if this would help improve the impact of the full SCDA. This will be an important future study. Further, one possible reason why $\mathbf{A}_\times^\times \mathbf{L}_\times^\times$ (SCDA) did not always show optimal results in the current study could be due to the poor and complex physical linkages between the lower troposphere and soil moisture. This problem has reasonably positive effects on the atmospheric field but often results in poor soil moisture analysis. The results presented in this study seem to indicate that this may be the case for $\mathbf{A}_\times^\times \mathbf{L}_\times^\times$ (SCDA).

Assimilating SM decreased the temperature estimates for the lower troposphere in areas with a dry SM bias and a warm atmospheric bias. This effect occurred because more incoming solar and longwave radiation was converted to latent heat flux and less converted to sensible heat flux with increased SM. However, assimilating SM into atmospheric model variables led to overcooling effects in regions such as the continent of Africa and Australia. Furthermore, estimating precipitation based on SCDA was beneficial in Africa. Coupled DA had stronger impacts on precipitation forecasts in hotspots where the

precipitation field is sensitive to the initial condition of SM.

This study demonstrated the potential for improving SM prediction using the NICAM-LETKF system by assimilating SM in strong coupling. SM is an important variable in land surface models, and its improvement can lead to better hydrological predictions such as droughts and floods. However, it is still unclear what atmospheric variables should be updated using each land observation. Therefore, future studies will further investigate the effect of variable localization for other land

observations. When assimilating SM by updating it in MATSIRO with atmospheric observations, we obtained unfavorable results due to errors in estimating the error covariance between land model variables and atmospheric observations. The issue is thought to be caused by experimental settings, rather than statistical aspects, due to the poor physical relationships between the lower troposphere and SM. SM behavior is often highly localized due to spatial differences such as soil texture, topography, and vegetation. Therefore, most NWP centers use a point-wise analysis of SM, without considering the

horizontal background error covariance between grid points. The 40 ensemble members used in this study were close to the number used in operational NWP centers, but a larger number of ensembles could lead to useful conclusions by evaluating the differences in performance between WCDA and SCDA. Furthermore, using a large ensemble could be beneficial for understanding variable localization more accurately by improving covariance estimation between components. In this study, land observations were assimilated into the atmospheric model using the same vertical localization scale as the assimilation



of atmospheric observations. Using a smaller localization scale in a limited ensemble size could help update atmospheric variables with SM assimilation by reducing errors in the error covariance estimates. Furthermore, while this study used SM data based on GLDAS, assimilating satellite-derived SM data is an important direction for future research. When actual GCOMW/AMSR-2 satellite observation data were assimilated, the atmospheric field deteriorated significantly due to the assimilation of SM (not shown). This suggests that limitations exist in the data assimilation method used in this study and

that technical measures such as CDF matching preprocessing be necessary to assimilate actual observation data successfully. Finally, it was found that resolution of about 100 km is very coarse to simulate SM accurately. When actual observation data is assimilated at this resolution, the representation error becomes large and will cause a problem. In addition to using actual satellite observation data, using higher-resolution models is an important future direction.

**Competing interests**

At least one of the (co-)authors is a member of the editorial board of Nonlinear Processes in Geophysics.

**Acknowledgments.**

K. Kurosawa and S. Kotsuki developed the experimental system for the parameter estimation, conducted the experiments and analyzed the results. T. Miyoshi is the PI and directed the research with substantial contribution to the development of this paper.

The authors thank the members of Data Assimilation Research Team, RIKEN Center for Computational Science (R-CCS) and JAXA's PMM project. This study was partly supported by JAXA Precipitation Measuring Mission (PMM), Advancement of meteorological and global environmental predictions utilizing observational 'Big Data' of the social and scientific priority issues (Theme 4) to be tackled by using post K computer of the FLAGSHIP2020 Project of the Ministry of Education, Culture, Sports, Science and Technology Japan (MEXT), the Initiative for Excellent Young Researchers of

MEXT, JST AIP Grant Number JPMJCR19U2, the Japan Society for the Promotion of Science (JSPS) KAKENHI grant JP18H01549 and 21H04571, JST PRESTO MJPR1924, and IAAR Research Support Program of Chiba University. The study used the Supercomputer for earth Observation, Rockets, and Aeronautics (SORA) at JAXA, and the K computer provided by the RIKEN R-CCS (Project IDs: ra000015, hp150289, hp160229, hp170246, and hp180062).

**Data Availability Statement**

The NICAM model code is available at http://www.nicam.jp/. The GSMaP precipitation data is available at http://sharaku.eorc.jaxa.jp/GSMaP/. The NCEP PREPBUFR data is available at http://rda.ucar.edu/datasets/ds337.0/. The GLDAS soil moisture data is available at https://hydro1.gesdisc.eosdis.nasa.gov/data/GLDAS/. The GPCP precipitation data is available at http://eagle1.umd.edu/GPCP_CDR/. The SMOS data is available at https://smos-diss.eo.esa.int/oads/access. The GCOMW/AMSR-2 data is available at https://lance.nsstc.nasa.gov/amsr2-science. The LETKF code developed in this

study is based on the open source code available at https://github.com/takemasa-miyoshi/letkf. All of the data used in this study are stored for 5 years in Chiba University. Due to the large volume of data and limited disk space, data will be shared online upon request (shunji.kotsuki@chiba-u.jp; http://www.data-assimilation.riken.jp/index_e.html).

**APPENDIX**

This study diagnoses the observation error SD of SM by using innovation statistics (Desroziers et al., 2005). The innovation

statistics is given by:



$$(\sigma^o_{estimation})^2 = \langle(y^o - H\overline{x}^a)(y^o - H\overline{x}^f)\rangle, \tag{A1}$$

where, $\sigma^o$ is the observation error SD. Subscript *estimation* means the estimation by the innovation statistics. The bracket $\langle\cdot\rangle$ denotes the statistical expectation. Here, we assumed the observations error SD is globally constant and time independent for SM (Rodríguez-Fernández et al., 2019). With NICAM-LETKF, we performed preliminary WCDA and SCDA experiments over two months from October to November 2014, and used later one month period data for the innovation statistics. Here we introduce a measure *factor,* given by

$$Factor = \sigma^o_{estimation}/\sigma^o_{prescribed}, \tag{A2}$$

where the subscript *prescribed* denotes the prescribed observation error SD of SM used in the preliminary experiments. If the prescribed SD is optimal, then the diagnosed *factor* approaches 1.0. Table A1 summarizes the prescribed observation error SD and *factor* values for five different observation SDs with assimilation of GLDAS SM. As noted by Ménard et al. (2009), when the prescribed observation error SD is too small, the estimated observation error SD is underestimated, whereas large SDs can lead to overestimation. Based on these preliminary experiments, this study set the SM observation error SD at 0.05 $(m^3\,m^{-3})$, which gave the *factor* value closest to 1.0 among the preliminary experiments.

**Table A1. Observation error SD diagnosed using innovative statistics. "Factor" is the ratio of estimated error SD to the prescribed value (Eq. A2). The diagnostic values from $A^\times_- L^-_\times$ (WCDA) and $A^\times_\times L^\times_\times$ (SCDA) averaged over 2 months are shown.**

| Prescribed Obs. Error SD $(m^3\,m^{-3})$ | Factor | |
|---|---|---|
| | $A^\times_- L^-_\times$ (WCDA) | $A^\times_\times L^\times_\times$ (SCDA) |
| 0.01 | 9.263 | 5.716 |
| 0.03 | 1.881 | 1.515 |
| 0.05 | 0.898 | 0.775 |
| 0.07 | 0.543 | 0.509 |
| 0.09 | 0.373 | 0.359 |



710

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
