# Peer review of "Comparative Study of Strongly and Weakly Coupled Soil Moisture Data Assimilation with a Global Coupled Land-Atmosphere Model"

_EGUsphere, 2023_

## Author Comment (AC1)

**Replies to the reviewers' comments**

Kenta Kurosawa, Shunji Kotsuki, and Takemasa Miyoshi

August 2, 2023

Manuscript No.: egusphere-2023-887

Title:

→

Title: "Comparative Study of Strongly and Weakly Coupled Data Assimilation for a Global Land-Atmosphere Coupled Model"

Thank you very much for your favorable evaluation to our work and insightful review comments. The followings are our point-by-point replies to the comments. Our replies are indicated by the blue font, and the reviewers' line number of the corresponding responses may be changed because of the revision. Supplemental PDF file would be useful to check revisions and their corresponding comments.

================================================================================

**[Reviewer 1]**

**General Comments**

This article has made valuable contributions by investigating the strong coupled data assimilation (SCDA) of land and atmospheric models. The experiments conducted in this study examined the effectiveness of SCDA in improving the simulation of soil moisture in land models using multiple observational data. The paper has yielded useful findings, e.g., analyzing atmospheric variables by assimilating SM data improves SM analysis and forecasts and mitigates a warm temperature bias in the lower troposphere where a dry SM bias exists; whereas analyzing SM by assimilating atmospheric observations has detrimental impacts on SM analysis and forecasts. These results hold significant implications for future improvements in numerical modeling and forecasting.

The coupled models and data assimilation system employed in this paper build upon previous work by the research team, providing a solid foundation. The experimental results demonstrate significant improvements in the representation of soil moisture variables through strong coupling assimilation of soil moisture data. The paper effectively elucidates the reasons behind these improvements through comparative analysis. Additionally, it highlights the adverse effects and their underlying causes when assimilating atmospheric data for SCDA of soil moisture variables. The organization of the paper is

reasonable, and the presentation is clear. The appropriate use of figures further enhances the interpretation of the results.

The following are some comments based on my personal observations.

Reply:

Thank you very much for the careful review. We revised the manuscript accordingly following the suggestions.

**Major comments**

(1) The title of the paper, "Comparative Study of Strongly and Weakly Coupled Soil Moisture Data Assimilation with a Global Coupled Land-Atmosphere Model," does not accurately reflect the comprehensive discussion presented in the paper. While the paper does investigate both strongly and weakly coupled data assimilation within the Coupled Land-Atmosphere Model, assimilating both atmospheric and land observations. Furthermore, the paper conducts experiments specifically examining the SCDA/WCDA on SM variable with atmospheric observations and discusses the results. Therefore, the title appears to be somewhat inadequate in capturing the full extent of the paper's content.

Reply:

We have revised the title as follows:

"*Comparative Study of Strongly and Weakly Coupled Data Assimilation for a Global Land-Atmosphere Coupled Model*"

(2) The paper introduces multiple scenarios to compare the strong/weak coupling assimilation of different model variables with different observational data. Although the authors employ their defined IDs (such as A^x_xL^x_-) to represent these scenarios, as depicted in Figure 2, the representation of these IDs is not particularly user-friendly. The correspondence between the "yes" and "no" labels in Figure 2 and the "x" and "-" symbols in the IDs is unclear. Additionally, the order of A and L in Equation (8) and the naming of cases (c - f) as "one-way SCDA" scenarios may lead to confusion. I suggest using simpler expressions. For reference, (e) could be referred to as "AtoL SCDA", and (f) as "LtoA SCDA". Referring to Penny and Hamill (2017), case (c) wich strongly assimilated only atmospheric observations could be considered quasi-SCDA, while the CTRL scenario would be quasi-WCDA. The (d) scenario is quite distinctive, representing the assimilation of atmospheric variables using both atmospheric and land observations. Therefore, describing it as one-way SCDA would be inaccurate, and a more appropriate abbreviation is required. By adopting more suitable and concise terminology, the reliance on cumbersome IDs can be minimized, leading to enhanced readability.

Reply:

We have changed the IDs in accordance based on your suggestion and the naming convention

of Penny and Hamill (2017).

(a) $A_{A\times}L_{\times\times}$ (CTRL; quasi-WCDA)
(b) $A_{A\times}L_{\times L}$ (WCDA)
(c) $A_{A\times}L_{A\times}$ (quasi-SCDA)
(d) $A_{AL}L_{\times\times}$ (quasi-SCDA)
(e) $A_{A\times}L_{AL}$ (SCDA)
(f) $A_{AL}L_{\times L}$ (SCDA)
(g) $A_{AL}L_{AL}$ (Full-SCDA)

We provided additional sentences about IDs to address the reviewer's comment (Sec. 2b, Fig. 2).

(3) The paper assimilates reanalysis data from GLDAS, and the validation data used for comparison also comes from GLDAS. However, the authors' explanation of GLDAS is not sufficiently detailed. It is unclear whether GLDAS incorporates results from multiple models, and if so, whether the authors only assimilate a subset of those models while using the rest for validation. The mention of "Noah model-based SM data" on line 224 suggests that it could be one of the models within GLDAS. However, it remains uncertain whether GLDAS includes results from other models alongside Noah.

Reply:

GLDAS employs a variety of land surface models (LSMs), among which includes the Noah model. Each LSM simulates the state of the land surface based on differing physical dynamics. In this study, we utilize the Noah model. This model is a general one for representing land surface processes including soil, evapotranspiration, precipitation, vegetation, and snowpack. We provided additional sentences about observations to address the reviewer's comment. (Sec. 2d)

Furthermore, in this study, we assimilate data from GLDAS. To verify whether our data assimilation system is functioning properly, we compare our results with GLDAS in Figures 4-8. After that, particularly from Sec. 4b, we use data from ERA5 and precipitation to perform validations not only on soil moisture but also on atmospheric variables.

(4) In Section 2.2, the authors briefly describe how coupled data assimilation is implemented using the LETKF method. However, this description does not apply to all seven scenarios. In Section 2.2, Equation (8) is used to represent the covariance of the entire system, suggesting that x includes variables from both components. However, in scenarios (a) and (d), x should not include the land component, as it would result in a singular matrix for the covariance P. This should be

clarified somewhere in the paper.

Reply:

You're right in pointing out that in scenarios (a) and (d), the variable x should not include the land component, which our current explanation may have confused. We added descriptions to address the reviewer's comment (Section 2.2)

"Note that the state variable x^f does not include the land component when the land variables are not updated (cf. Figs. 2 a and d). For such cases, the forecast error covariance matrix also has the inverse matrix since the land component is also excluded in the background error covariance."

(5) I am not entirely clear about the purpose of Figure 3. Does it explain the rationale behind assimilating reanalysis data?

Reply:

We employ SM data from GLDAS, a comprehensive and reliable dataset which facilitates simple data handling and is suitable for this study (cf. Section 2d). Figure 3 is used to demonstrate that assimilation of real satellite observation data requires preprocessing, including quality control and bias correction, prior to assimilation.

(6) The reference cited in line 336, Kang et al. 2011, is not listed in the final bibliography.

Reply:

We have added "Kang, J.-S., E. Kalnay, J. Liu, I. Fung, T. Miyoshi, and K. Ide, 2011: "Variable localization" in an ensemble Kalman filter: Application to the carbon cycle data assimilation. J. Geophys. Res., 116, D09110, https://doi.org/10.1029/2010JD014673."

(7) The statement made in line 429 suggests that the advantages of SCDA increase as the peak of precipitation moves northward. However, the results may not demonstrate the extent of this effect as explicitly as indicated by the authors. Could this be quantified using relevant numerical measures, such as the difference in regional average RMSE?

Reply:.

Figure 10 aims at observing seasonal differences across different regions. While we understand that the numerical quantification using regional average RMSE could provide more precise insights, the process to compute RMSE for each region is quite tedious. We believe that the current representation is sufficiently illustrative to aid understanding without the need for explicit numerical quantification through RMSE.

With regard to the clarity of our statement in line 429, I agree that the initial wording might have been ambiguous. Therefore, we have revised the text in the mentioned line to explicitly state the relationship between the advantages of SCDA and the movement of the peak of precipitation

northward. Your suggestion to quantify the results with numerical measures is appreciated, and we have considered it carefully in our revisions. (lines 545-569)

(8) Figure 12 shows that the difference between strong and weak coupling is not significant when compared to ERA5, as long as both observations are assimilated. Does this indicate that the earlier results heavily rely on using the same data for assimilation and validation?

Reply:

The primary purpose of Figure 12 is to show that experiment $A_{AL}L_{\times\times}$ (quasi-SCDA; Fig. 12d ) demonstrates better scores than $A_{A\times}L_{\times\times}$ (CTRL), while experiment $A_{A\times}L_{A\times}$ (quasi-SCDA; Fig. 12c) exhibits poorer scores.

We provided additional sentences about the points to address the reviewer's comment (Fig. 12, lines 601-604).

The above are just some of my questions and suggestions regarding the paper. Overall, the work conducted in the paper is meticulous and comprehensive. The authors have also provided detailed discussions and future prospects. I support the publication of this paper and hope that the authors will consider my suggestions for minor revisions.

===================================================================================

**[Reviewer 2]**

**General comments**

In this study, the authors investigated the effect s of weakly and strongly-coupled DA experiments by assimilating atmospheric observations and SM data simultaneously using a coupled land-atmosphere model. The topic of this paper is interesting, which is expected to provide more evidence for improving NWP by the land-atmosphere DA. However, some detailed descriptions about the DA experiments are missing. Therefore, I would recommend this work for publication after minor revisions. Here are my detailed comments.

Reply:

Thank you very much for the careful review. We revised the manuscript accordingly following the suggestions.

**Major comments**

(1) My main criticism is how to calculate the cross-component error covariance between atmosphere and land variables in the strongly coupled DA (Eq. 8). How to examine the impact of the

generation of ensemble samples on the cross-component error covariance or assimilated results?

Reply:

You have raised an essential point about the calculation of the cross-component error covariance between atmospheric and land variables in the strongly coupled DA (Eq. 8). We understand now that the concern may lie in understanding how the background error covariance is constructed. We added explanations on this point (Line 158-164).

(2) Abstract: More detailed conclusions are missing in the abstract.

Reply:

I have revised it to include more detailed conclusions.

(3) L280-285: Details about the soil moisture state variables are needed. Since the top two soil layers of MATSIRO are 0-0.05 and 0.05-0.25 meters while 0-10cm for GLDAS, how to set the soil moisture state variables?

Reply:

In our experiments, GLDAS SM data are assimilated into the topmost layer of MATSIRO ($0-0.05$ meters). We added descriptions to address the reviewer's comment. (Line 250-270, Line 558-587).

(4) L256: What is RTPS?

Reply:

Thank you for your query. RTPS stands for 'Relaxation to Prior Spread', as explained in the preceding paragraph of the manuscript.

(5) Updating SM by assimilating atmospheric observations had detrimental impacts on SM due to spurious error correlations between atmospheric observations and land model variables. However, the error correlation between SM observations and atmospheric model variables is more reliable when updating atmospheric model variables by assimilating SM observations. It is hard to understand this conclusion. More detailed interpretations are needed.

Reply:

We explained that in more detail in Section 4a (Figure 5) with the following sentences.

"The larger error in $A_{A \times} L_{AL}$ (SCDA) than in $A_{A \times} L_{\times L}$ (WCDA) arises from inaccurate covariance estimates between atmospheric observations and land variables due to insufficient ensemble size. Ensemble-based DA can provide spurious error correlations when the ensemble size is small. Assimilating observations based on spurious error covariances generally degrades the analysis results (cf. variable localization of Kang et al. 2011).

Moreover, the difference in timescale between the atmospheric and terrestrial models may have a dominant influence, which could be verified by experiments using a short assimilation window. Such further investigation of the assimilation window is essential for future studies of land-atmosphere coupled DA."

We added descriptions to address the reviewer's comment (Sec. 5; Lines 804-812).